# Cooperation between oncogenic Ras and wild-type p53 stimulates STAT non-cell autonomously to promote tumor radioresistance

Yong-Li Dong[1,2,8], Gangadhara P. Vadla[3,8], Jin-Yu (Jim) Lu[1,4], Vakil Ahmad[3], Thomas J. Klein[1,5], Lu-Fang Liu[1], Peter M. Glazer [6], Tian Xu[1,7✉] & Chiswili-Yves Chabu [3✉]

Oncogenic RAS mutations are associated with tumor resistance to radiation therapy. Cell-cell interactions in the tumor microenvironment (TME) profoundly influence therapy outcomes. However, the nature of these interactions and their role in Ras tumor radioresistance remain unclear. Here we use *Drosophila* oncogenic Ras tissues and human Ras cancer cell radiation models to address these questions. We discover that cellular response to genotoxic stress cooperates with oncogenic Ras to activate JAK/STAT non-cell autonomously in the TME. Specifically, p53 is heterogeneously activated in Ras tumor tissues in response to irradiation. This mosaicism allows high p53-expressing Ras clones to stimulate JAK/STAT cytokines, which activate JAK/STAT in the nearby low p53-expressing surviving Ras clones, leading to robust tumor re-establishment. Blocking any part of this cell-cell communication loop re-sensitizes Ras tumor cells to irradiation. These findings suggest that coupling STAT inhibitors to radiotherapy might improve clinical outcomes for Ras cancer patients.

[1] Howard Hughes Medical Institute, Department of Genetics, Yale University School of Medicine, Boyer Center for Molecular Medicine, New Haven, CT, USA. [2] State Key Laboratory of Genetic Engineering and National Center for International Research, Fudan-Yale Biomedical Research Center, Institute of Developmental Biology and Molecular Medicine, School of Life Sciences, Fudan University, Shanghai, China. [3] Division of Biological Sciences, College of Veterinary Medicine, Department of Surgery, University of Missouri, Columbia, MO, USA. [4] Yale-Waterbury Internal Medicine Residency Program, Waterbury, CT, USA. [5] South Florida Radiation Oncology, West Palm Beach, FL, USA. [6] Department of Therapeutic Radiology, Yale University School of Medicine, New Haven, CT, USA. [7] Key Laboratory of Growth Regulation and Translation Research of Zhejiang Province, School of Life Sciences, Westlake University, Hangzhou, Zhejiang Province, China. [8] These authors contributed equally: Yong-Li Dong, Gangadhara P. Vadla. ✉email: tian.xu@yale.edu; chabuc@missouri.edu

Oncogenic *Ras* mutations activate a complex network of interacting signals to cause aggressive cancers[1,2]. Gold standard treatment options include radiation therapy and conventional chemotherapies that cause irreversible genomic damage and trigger apoptosis[3]. However, oncogenic Ras mutations enable cancer cells to resist these genotoxic agents, ultimately leading to cancer recurrence[4–13]. We define tumor radioresistance as incomplete sensitivity and/or the capacity of tumors to rapidly re-form following radiation therapy.

Various mechanisms have been proposed to explain the resistance of *Ras* cancers to treatments, including the presence of cancer stem cells in the tumor microenvironment[5,14–16]. Another view is that therapy-resistant cancer cells possess robust DNA repair mechanisms that curtail the proapoptotic effect of the treatment. In many cancers, including lung and colorectal cancers where oncogenic Ras mutations are common, an association between polymorphisms in DNA damage response genes and an improved clinical response to genotoxic agents has been observed[17–23]. However, cellular responses to DNA damage are complex and include activation of cell–cell interactions that we do not fully understand[24]. How these nonautonomous effects influence the response of Ras-driven cancers to genotoxic therapies is an underexplored area of research. Animal tumor models provide the advantage of interrogating tumor resistance mechanisms at the tissue level, enabling the identification of novel and broadly applicable mechanisms.

In genetic screens for suppressors of oncogenic *Ras* (*Ras^{V12}*)-mediated tissue overgrowth in *Drosophila*[25,26], we isolated genotoxic mutations, including null alleles of the Pax2 transactivation domain-interacting protein coding gene (*ptip^{−/−}*). Interestingly, *ptip^{−/−}* inhibits the growth of *Ras^{V12}* cells but also triggers the overgrowth of the surrounding tissues. PTIP is essential for maintaining genomic stability under normal conditions and after DNA damage[27–29]. Disruption of the PTIP DNA repair complex causes genomic instability and triggers a DNA damage response that culminates in the activation of *p53* (*dp53* in *Drosophila*), which orchestrates DNA repair or triggers apoptosis of damaged cells[29,30].

It is becoming evident that p53 biology is far more complex than initially thought and involves nonautonomous functions that are not well-understood[31–34]. We found that *ptip^{−/−}* causes genomic instability and consequently upregulates *dp53* in *Ras^{V12}* cells. This upregulation of wild-type *dp53* cooperates with oncogenic *Ras* signaling to stimulate the secretion of JAK/STAT (Janus kinases/signal transducers and activators of transcription) ligands (interleukin 6-related cytokines known as unpaired in *Drosophila*). These ligands activate JAK/STAT in the surrounding cells, leading to tissue overgrowth. Ionizing radiation (IR) of *Drosophila Ras^{V12}* tissues or of human cancer cells harboring oncogenic *Ras* mutations triggers similar nonautonomous effects. Blockade of any part of this p53/Ras^{V12}-STAT signaling relay inhibits the nonautonomous growth effect and resensitizes *Ras^{V12}* tissues to IR treatment.

In addition to highlighting the complexity of p53 biology, our work defines a treatment-induced cell–cell interaction dynamic that promotes the recurrence of oncogenic Ras mutant tumors after genotoxic therapies. Our data also provide a possible explanation for why some Ras mutant cancers resist genotoxic therapies despite the lack of p53 mutations.

## Results

### *Ptip^{−/−}* promotes nonautonomous tissue overgrowth.
In *Drosophila*, the MARCM (mosaic analysis with a repressible cell marker) technique permits the expression of oncogenic Ras (*Ras^{V12}*) in clones of cells within the developing eye epithelium[26].

These clones coexpress a fluorescent protein, making them distinguishable from the surrounding wild-type cells. *Ras^{V12}*-mediated tissue overgrowth is readily detectable by the appearance of large and hyperplastic fluorescent clones (tumors) that ultimately kill the animal[35–37]. *Ras^{V12}* suppressor mutations are isolated through the identification of mutations that significantly reduce the clone size and rescue animal viability when introduced in *Ras^{V12}*-expressing cells[35].

We isolated several *Ras^{V12}* suppressors, including mutation #3804, using this approach. Mutation 3804 potently suppresses *Ras^{V12}*-mediated tumor overgrowth and yields viable adult animals (Fig. 1a versus 1b; 1c versus 1d; 1g–l). To determine whether this mutation synthetically suppresses oncogenic Ras or is cell deleterious on its own, we generated wild-type or 3804 mutant clones in developing and adult eye tissues to determine whether this mutation synthetically suppresses oncogenic Ras or is deleterious to the cell itself. Adult eye clones are marked by the absence of the red pigment. As expected, wild-type cells contributed ~50% to the eye field. In contrast, 3804 mutant cells were barely detectable in tissues, suggesting that the affected gene is essential for cell viability (Fig. 1m–o).

Interestingly, although 3804 suppressed *Ras^{V12}* tumor overgrowth in the developing eye tissue and correspondingly yielded adult animals, the eyes of the rescued animals were overgrown to varying extents, as evidenced by the appearance of tissue folds (Fig. 1e, f versus 1g–l, w). The overgrown tissues exclusively consisted of wild-type cells (GFP-negative) (Fig.1p–s).

We used deficiency mapping and allele sequencing approaches and determined that 3804 represents a null mutation in the PAX transcriptional activation domain-interacting protein (PTIP). Homozygosity of the 3804 mutation is animal lethal. Two independent deficiency alleles (ED4529 and ED4536) that overlap at the PTIP gene fail to complement 3804 animal lethality (Fig. 1t). Direct sequencing of the 3804 allele revealed a G>A mutation leading to a premature stop codon in the protein (Fig.1u, v). Thus, we hereafter refer to 3804 as *ptip^{−/−}* and conclude that while *ptip^{−/−}* suppresses growth in a cell-intrinsic manner, it cooperates with oncogenic Ras to promote the growth of the surrounding tissue.

### *Ptip^{−/−}* promotes nonautonomous tissue overgrowth via p53.
We sought to delineate the underlying mechanism. PTIP was originally identified in a yeast two-hybrid screen[38]. In mammals, PTIP interacts with histone methyltransferase complexes to control developmental transcription programs[38,39]. Additionally, PTIP is essential for maintaining genomic stability[29,40,41]. Genomic instability triggers the DNA damage response signals, which culminate in the sequential activation of p53 and p21 (Dacapo or dap in *Drosophila*) to drive cell cycle arrest or apoptosis[42–44]. We investigated whether *ptip^{−/−}* causes DNA damage in *Drosophila Ras^{V12}* epithelial cells and found that it does. DNA damage triggers the phosphorylation of a histone 2A variant (γH2Av), which is readily detected in immunostaining experiments using phospho-specific antibodies[45,46]. We examined γH2Av in *ptip^{−/−}* cells or in *Ras^{V12}* cells with or without the *ptip^{−/−}* mutation. We observed an increase in the number of γH2Av nuclear foci in *ptip^{−/−}* and *Ras^{V12}ptip^{−/−}* double mutant cells, but not in cells expressing *Ras^{V12}* alone (Fig. 2a–c'). In addition, quantitative polymerase chain reaction (qPCR) assays revealed transcriptional upregulation of *dp53* and *dap/p21* in *ptip^{−/−}* or *Ras^{V12}ptip^{−/−}* double mutant tissues compared to wild-type controls (Fig. 2j, k). Consistent with DNA damage, *ptip^{−/−}* caused an increase in dp53 and dap/p21 protein levels in *Ras^{V12}* cells (Fig. 2d–g'). We posited that the cellular response to genomic stress likely underlies the nonautonomous growth effect of *ptip^{−/−}* on the

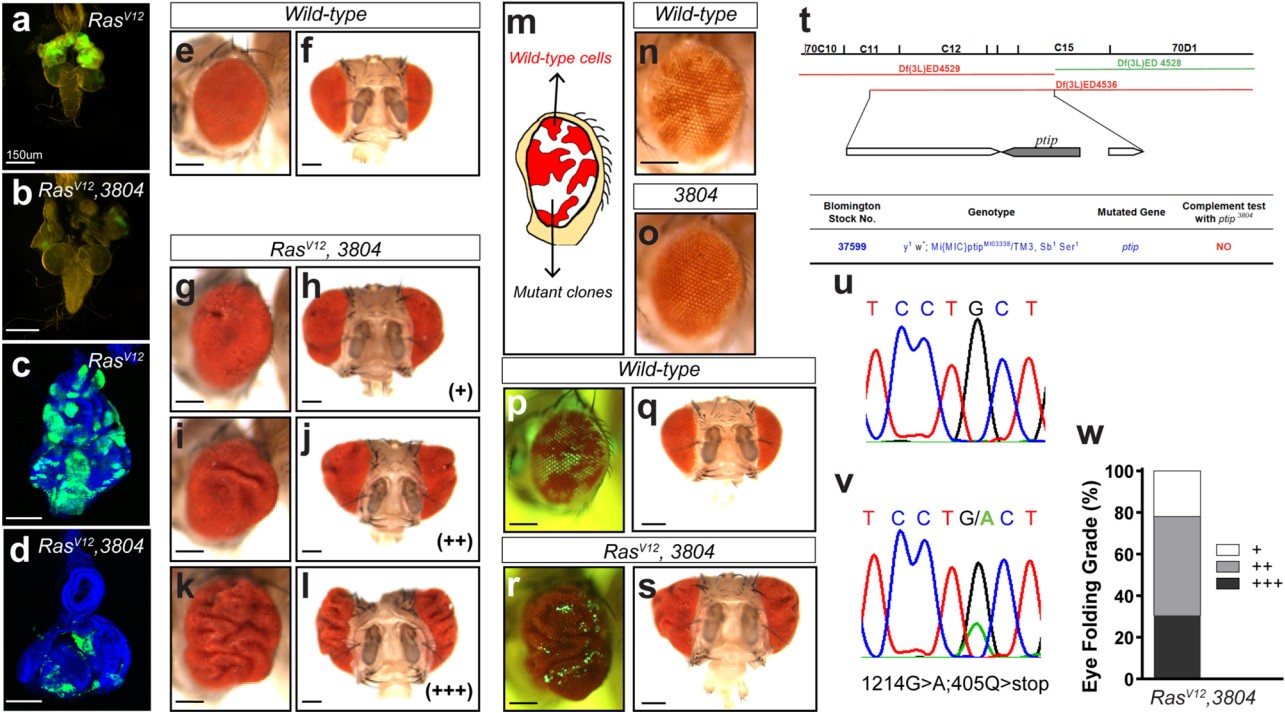

**Fig. 1 Ptip$^{-/-}$ cooperates with oncogenic Ras to induce nonautonomous tissue overgrowth.** (**a–d**) Mutation *3804* suppresses *Ras$^{V12}$*-mediated tumor overgrowth. (**a, b**) Images of third-instar larvae cephalic complexes showing GFP-marked *Ras$^{V12}$* (**a**) or *Ras$^{V12}$3804* (**b**) clones. Images of eye imaginal discs dissected from third-instar larvae cephalic complexes and containing *Ras$^{V12}$* or *Ras$^{V12}$ 3804* GFP-positive clones are shown in (**c**) and (**d**), respectively. Scale bars are 150 μm. (**e–l**) Side and frontal images of adult fly eyes. The nonautonomous overgrowth phenotype was categorized into three grades based on the severity of the phenotype (+: weak; ++: moderate; +++: strong). (**e**) and (**f**) represent adult eyes from wild-type animals. The adult eye tissues bearing *Ras$^{V12}$ 3804* double mutant clones showed varying grades of tissue folding (**g–l**). (**m–o**) Mosaic adult eyes bearing wild-type cells and mutant clones marked by a lack of pigmentation. A schematic of the mosaic adult eye is presented in (**m**). Wild-type cells (*n*, white color) contribute ~50% to the eye field, whereas the *3804* mutant clones (**o**) are barely detectable. (**p–s**) Matched light and fluorescence images of adult eyes containing GFP-positive wild-type (**p, q**) or *Ras$^{V12}$ 3804* double mutant clones (**r, s**). (**t**) Genetic complementation test of the *ptip$^{-/-}$* mutation using overlapping chromosomal deficiency lines. The deficiency line shown in green complements the *ptip$^{-/-}$* mutation, while the deficiency lines marked in red fail to complement. (**u, v**) Sequence results showing a G > A mutation in *PTIP*, causing a premature stop sequence. (**w**) Quantification of (**g–l**). All scale bars are 150 μm.

*Ras$^{V12}$* clones. Janus N-terminal kinase (JNK, also known as Bsk in *Drosophila*) and p53 play central roles in cellular response to DNA damage[47–49]. In addition, we and other researchers have shown that JNK promotes nonautonomous tissue growth in *Drosophila*[35,50,51], suggesting that *ptip$^{-/-}$* acts via JNK to drive nonautonomous growth in *Ras$^{V12}$* mosaic tissues. Consistent with this hypothesis, JNK was activated in *Ras$^{V12}$ ptip$^{-/-}$* mutant cells compared to *Ras$^{V12}$* control cells (Fig. 2h-i′). We inhibited JNK by expressing a potent dominant-negative JNK transgene (*Bsk$^{DN}$*) in *Ras$^{V12}$ptip$^{-/-}$* cells (*Ras$^{V12}$ptip$^{-/-}$Bsk$^{DN}$* triple defective cells) and asked whether this genetic manipulation suppresses the nonautonomous tissue overgrowth phenotype to directly test this hypothesis. *Bsk$^{DN}$* failed to suppress *Ras$^{V12}$ptip$^{-/-}$* nonautonomous tissue overgrowth (Figs. 2l–n and 2l′-n′), making it unlikely that JNK plays a significant role in this phenomenon.

We explored alternative mechanisms. Nonautonomous growth-inducing clones (*Ras$^{V12}$ptip$^{-/-}$* mutant cells) showed higher levels of the wild-type p53 protein (p53$^{wt}$) than *Ras$^{V12}$* cells, which did not cause nonautonomous growth (Fig. 2d-e′). Because the *ptip$^{-/-}$* mutation occurs very early and is permanent, the resulting high p53$^{wt}$ protein levels (Fig. 2d, d′, j, and k) likely persist throughout the life of *Ras$^{V12}$* cells. Normally, p53$^{wt}$ has a high turnover rate[52]. We wondered whether the elevated p53$^{wt}$ protein levels observed in the *Ras$^{V12}$ptip$^{-/-}$* mutant cells play an active role in the nonautonomous tissue growth effect. Indeed, RNAi knockdown of dp53 in *Ras$^{V12}$ptip$^{-/-}$* cells remarkably suppressed nonautonomous tissue overgrowth (Fig. 2m, m′, o, o′, and s). Similarly,

blocking dp53 transcriptional activity in *Ras$^{V12}$ptip$^{-/-}$* cells by expressing a DNA binding-defective *dp53* mutant version (*p53$^{R155H}$*)[53] also suppressed the overgrowth of the surrounding wild-type cells (Fig. 2m, m′, p, p′, and s). In addition, direct overexpression of *p53$^{wt}$* (*p53$^{OE}$*) in clones of *Ras$^{V12}$* cells was sufficient to trigger the overgrowth of the surrounding wild-type tissue, mimicking *Ras$^{V12}$ptip$^{-/-}$* clones (Fig. 2l, l′, r, and r′). The ability of p53$^{wt}$ to drive nonautonomous tissue overgrowth requires oncogenic Ras. Overexpression of *dp53$^{wt}$* alone failed to generate a similar effect (Fig. 2l, l′, q, and q′).

We used the MARCM technique to juxtapose *Ras$^{V12}$p53$^{OE}$* clones (RFP-labeled) with *Ras$^{V12}$* clones (GFP-labeled) and assessed whether this *Ras$^{V12}$*/p53 cooperation similarly accelerates the growth of adjacent *Ras$^{V12}$* cells (see Methods). This alteration caused massive overgrowth of *Ras$^{V12}$* clones compared to controls (abutting *Ras$^{V12}$* clones without *dp53$^{OE}$*) (Fig. 2t-u″ and s). *Ras$^{V12}$ptip$^{-/-}$* clones exerted a similar nonautonomous effect on Ras clones (Supplementary Fig. 1). Taken together, these findings indicate that oncogenic Ras cooperates with elevated levels of the wild-type dp53 protein to drive tumor overgrowth via a novel nonautonomous mechanism.

**Oncogenic Ras and p53 cooperatively stimulate JAK/STAT cytokines to promote nonautonomous tissue overgrowth.** We set out to delineate the underlying mechanism. The *Drosophila* JAK/STAT ligands Unpaired1-3 (upd, upd2, and upd3) mediate

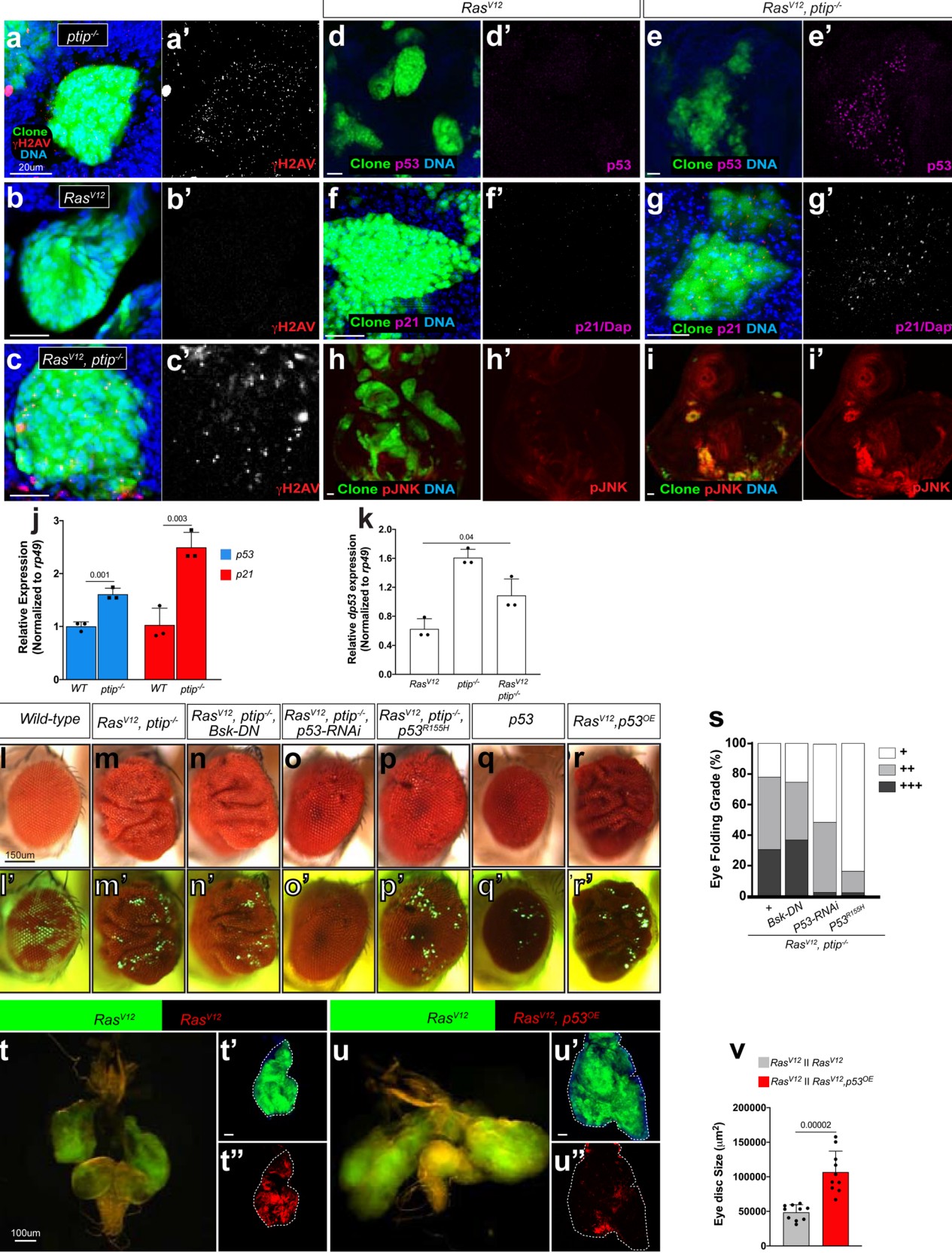

nonautonomous tissue growth[35,50,51]. We asked to what extent oncogenic Ras and dp53[OE] act via the JAK/STAT pathway. Immunostaining experiments using a *upd* reporter line, *upd-lacZ*[54], to monitor *upd* transcriptional activity revealed that

dp53[OE] causes *Ras*[V12] cells to upregulate *upd* (Fig. 3a–b'). In a complementary qPCR approach, we found that dp53[OE] causes *Ras*[V12] cells to upregulate all of the *upd* ligands (*upd1-3*) in tissues (Fig. 3d). The *ptip*[−/−] mutation exerted similar effects on *Ras*[V12]

**Fig. 2 $ptip^{-/-}$ and oncogenic Ras cooperatively induce nonautonomous tissue overgrowth via wild-type p53. (a–i')** Representative images of dissected eye imaginal discs containing $ptip^{-/-}$ or $Ras^{V12}$ or $Ras^{V12}ptip^{-/-}$ double mutant clones (GFP) stained with DAPI to detect DNA or anti-phosphorylated H2AV antibodies to detect DNA damage (**a–c'**) or anti-p53 (**d–e'**) or anti-dacapo (dap/p21) (**f–g'**) or anti-phosphorylated JNK (**h–i'**) antibodies to detect cellular response to DNA damage. Scale bars are 20 μm. (**j, k**) Quantitative Polymerase Chain Reaction (qPCR) data showing expression of p53 or dap/p21 in wild type versus $ptip^{-/-}$ eye imaginal discs (**j**) or the expression of p53 in $Ras^{V12}$ or $ptip^{-/-}$ or $Ras^{V12}ptip^{-/-}$ eye imaginal discs (**k**). Expression was normalized to the transcript abundance of the housekeeping gene rp49. Error bars denote standard deviation (SD) values. P values are derived from Student's t test analyses. (**l–r**) Matched light and fluorescence images of adult eyes containing GFP-labeled clones. The respective clone genotypes are indicated at the top of each panel. The corresponding fluorescent images are shown below in (**l'–r'**). GPF-negative tissues represent wild-type tissues. Scale bars are 150 μm. (**s**) Quantification of the nonautonomous growth phenotype of adult eyes containing clones of the indicated genotypes: $Ras^{V12}ptip^{-/-}$, $Ras^{V12}ptip^{-/-}Bsk^{DN}$, $Ras^{V12}ptip^{-/-}p53^{R155H}$, or $Ras^{V12}ptip^{-/-}p53^{RNAi}$. (**t–u"**) Genetic juxtaposition of GFP-labeled $Ras^{V12}$ clones with RFP-labeled $Ras^{V12}$ clones (**t–t"**, controls) or with RFP-labeled clones of cells coexpressing $Ras^{V12}$ and wild-type p53 ($Ras^{V12}$, $p53^{OE}$) (**u–u"**). GFP-positive $Ras^{V12}$ clones are surrounded by RFP/GFP double-positive $Ras^{V12}$ clones (**t–t"**) or by RFP/GFP double-positive $Ras^{V12}$, $p53^{OE}$ clones (**u–u"**). Brain cephalic complex images showing the growth of $Ras^{V12}$ clones when juxtaposed to $Ras^{V12}$ or to $Ras^{V12}$, $p53^{OE}$ clones are shown in **t** and **u**, respectively. Dotted white lines (**t', t", u', u"**) represent tissue boundaries. Scale bars are 100 μm. (**v**) Quantification of eye tissue sizes from (**t–u"**). Sample size N = 10 tissues per genotype. Error bars denote standard error of the mean (SEM) values. P values are derived from Student's t test analyses. Effect size (Cohen's d values) for (**j**), (**k**), and (**v**) is greater than 0.8.

tissues, including activation of JAK/STAT in cells surrounding the mutant ($Ras^{V12}ptip^{-/-}$) clones (Fig. 3a, a', c, c' and Supplementary Fig. 2a–d and 2g–j). We inhibited p53 via RNAi knockdown or by expressing dominant-negative $dp53$ ($p53^{R155H}$) in $Ras^{V12}ptip^{-/-}$ cells to further establish that $Ras^{V12}ptip^{-/-}$ tissues rely on dp53 for the stimulation of upd and found that each of these manipulations blocked the upregulation of upd ligands (Fig. 3e). These findings suggest that $Ras^{V12}p53^{OE}$ clones induce the growth of surrounding cells via the secretion of JAK/STAT cytokines. We blocked the secretion of JAK/STAT cytokines into the tissue surrounding $Ras^{V12}p53^{OE}$ clones and asked whether this blockade suppresses the nonautonomous growth effect to functionally test this hypothesis. We simultaneously knocked down upd and upd2 in $Ras^{V12}p53^{OE}$ clones by combining upd-RNAi expression with upd2 deletion mutants. This manipulation dramatically reduced the tissue size (Fig. 3f–h). Knockdown of Upd in $Ras^{V12}ptip^{-/-}$ clones similarly suppressed nonautonomous tissue overgrowth (Supplementary Fig. 2e, f, k, and l).

We evaluated our findings in human breast and lung cancer cells using supernatant transfer experiments. MCF-10A breast epithelial cells were cultured in media conditioned with MCF-10A cells (controls) or MCF-10A cells overexpressing wild-type p53 alone or coexpressing oncogenic HRAS. STAT signaling status was assessed using western blot experiments with antibodies that specifically detect activated STAT (anti-phosphorylated STAT). Growth was determined by scoring cell numbers. In this and subsequent experiments, the superscripts "P53OE" or "HRasG12V" denote overexpression of wild-type p53 or oncogenic Ras, respectively. As expected, MCF10A$^{HRasG12V, P53OE}$ cells showed elevated levels of the p53 protein and Ras signaling (determined by phospho-ERK) levels compared to untransfected controls (Fig. 4a, b). MCF10A$^{HRasG12V, P53OE}$-conditioned media stimulated STAT signaling and correspondingly induced the growth of MCF-10A cells (Fig. 4c, f, and supplementary Fig. 6). This growth-promoting effect was significantly reduced when the conditioning cells lacked oncogenic Ras (MCF-10A$^{P53OE}$) (Fig. 4f and supplementary Fig. 6).

Similar results were observed in lung cancer cells. We overexpressed wild-type p53 in H460, A549, H358, and H441 lung cancer cells, generating H460$^{P53OE}$, A549$^{P53OE}$, H358$^{P53OE}$, and H441$^{P53OE}$ cells (Fig. 4h and supplementary Fig. 6). Genetically, all of these cells carry oncogenic Ras mutations. However, endogenous p53 is wild-type in H460 and A549 cells and mutated in H358 and H441 cells[55]. Compared to matched controls, media conditioned with H460$^{P53OE}$, A549$^{P53OE}$, H358$^{P53OE}$, and H441$^{P53OE}$ cells stimulated STAT signaling and cell growth (Fig. 4i–k and supplementary Fig. 6). Media

conditioned with irradiated breast (MCF-10A$^{HRasG12V}$) or lung (H358, H460, and A549) cancer cells generated similar nonautonomous effects (Fig. 4g, l, n, and supplementary Fig. 6). Importantly, blocking STAT signaling with the small-molecule inhibitor Ganetespib[56] suppressed the nonautonomous growth-inducing effect of p53$^{OE}$ and IR on both breast and lung cancer cells (Fig. 4f, g, k, l, and supplementary Fig. 6).

Moreover, we tested our findings in vivo by performing mouse xenograft experiments. We inoculated one million A549 cells into each of the flanks of nude mice. Left flank inoculants were unmodified, while right flank inoculants consisted of a 50-50 mixture of untreated and irradiated cells. Eight weeks after treatment, tumor xenografts from the mixed population (right flanks) grew markedly larger than xenografts arising from the homogenous inoculants (left flanks) in the same animal (Fig. 4j and k; 83%; N = 6 animals). We treated animals with the validated pharmacological STAT blocker Ruxolitinib to test whether STAT plays a role in the observed tumor overgrowth, as suggested by our tissue culture and fly data[57]. Animals were treated orally with Ruxolitinib at 10 mg/kg, a dose that is well tolerated in nude mice[57]. Ruxolitinib suppressed the overgrowth of tumors arising from the mixed cells (Fig. 4p, q). Endpoint western blot analyses of tumor xenografts harvested from animals that were treated with or without Ruxolitinib confirmed the inhibition of STAT signaling in the treated and slower growing mixed tumors (Fig. 4r).

JAK/STAT signaling supports tissue growth by promoting cell survival or cell proliferation[58,59]. We examined cell death and cell proliferation in human cells using flow cytometry approaches to distinguish between these two mechanisms. Notably, p53$^{OE}$ and IR-induced nonautonomous JAK/STAT signaling mainly stimulated cell proliferation (Supplementary Figs. 3, 4).

Taken together, the above data indicate that stimulation of wild-type p53 cooperates with oncogenic Ras to induce JAK/STAT signaling in the surrounding cells, resulting in nonautonomous growth.

**The Oncogenic Ras/p53-STAT signaling relay promotes the radioresistance of *Drosophila* $Ras^{V12}$ tumor tissues.** In several cancer types, oncogenic Ras mutations are associated with disease recurrence following radiation therapy and genotoxic chemotherapies[4–13].

We used *Drosophila* $Ras^{V12}$ tumor tissues exposed to IR to test whether IR-stimulated dp53 generates similar growth-promoting effects in tissues. Specifically, we asked whether dp53 cooperates with oncogenic Ras to establish tumor recurrence via the nonautonomous STAT signaling relay described above.

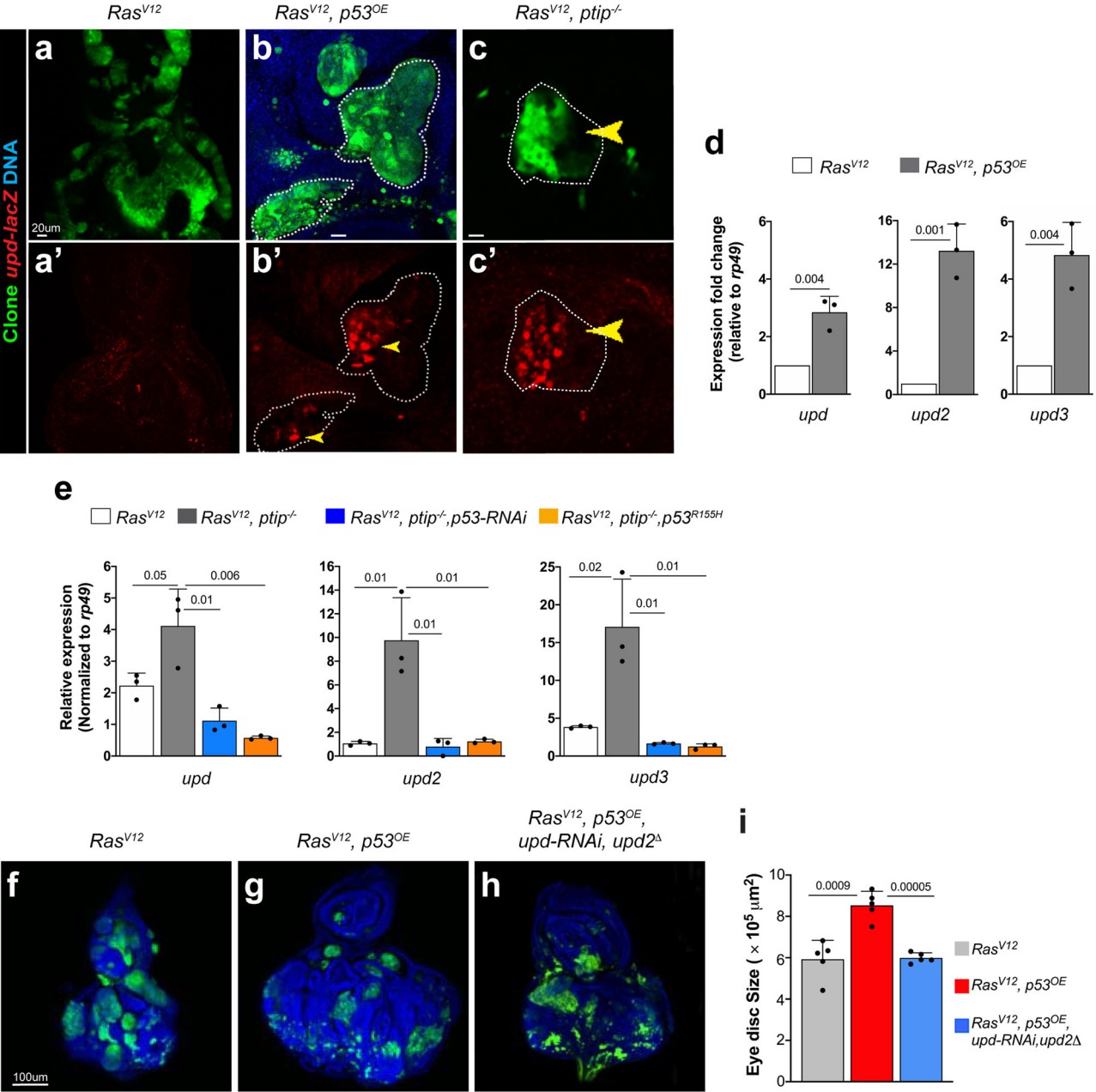

**Fig. 3 Wild-type p53 and oncogenic Ras cooperatively stimulate STAT cytokines to drive nonautonomous tissue overgrowth.** (**a–c'**) Images showing *upd-lacZ* expression in eye imaginal discs bearing $Ras^{V12}$ (**a, a'**), $Ras^{V12}p53^{OE}$ (**b, b'**), and $Ras^{V12}ptip^{-/-}$ (**c, c'**) clones. Anti-$\beta$gal antibodies were used in immunostaining experiments to detect LacZ. The individual LacZ channel images are shown in **a'–c'**. The dotted white lines denote clone boundaries and depict representative examples of *upd* overexpression (yellow arrowheads). Scale bars are 20 µm. (**d, e**) qPCR data showing expression of *upd*, *upd2*, and *upd3* in $Ras^{V12}$ versus $Ras^{V12}p53^{OE}$ (**d**) or $Ras^{V12}ptip^{-/-}$ in the absence or presence of p53 inhibition ($Ras^{V12}ptip^{-/-}p53^{RNAi}$ or $Ras^{V12}ptip^{-/-}p53R^{155H}$). Expression was normalized to the transcript abundance of *rp49*, a housekeeping gene. Error bars denote SD values. *P* values are derived from Student's *t* test analyses. (**f–h**) Representative images showing overall tissue size of dissected eye imaginal discs harboring GFP-labeled $Ras^{V12}$ (**f**) or $Ras^{V12}p53^{OE}$ (**g**) or $Ras^{V12}p53^{OE}upd^{RNAi}upd2^{\Delta}$ (**h**) clones. Blue signal represents DNA (DAPI stain). Scale bar is 100 µm. (**i**) Quantification of tissue sizes from (**f–h**). Sample size N = 06 tissues per genotype. Error bars denote SEM values. *P* values are derived from Student's *t* test analyses. Effect size (*d*) values for (**f–h**) are greater than 0.8.

We first performed a study to determine a dose that generates cellular effects without grossly impeding animal development. Second-instar larvae harboring GFP-labeled oncogenic Ras clones in eye imaginal discs were treated with 600 R, 1000 R, or 2000 R. Each dose was administered three times in 6 hours intervals to mimic clinical settings where total radiation treatments are administered in fractions[60,61]. Larvae treated with the 3 × 600 R dose developed normally into adults without any detectable

abnormalities, and those treated with 3 × 2000 R died during the pupal stage. Larvae that received 3 × 1000 R yielded adult flies with mild rough eyes, making it an ideal dosing regimen for our study.

We next determined the extent to which IR recapitulates fundamental aspects of the $Ras^{V12}/p53^{OE}$-STAT signaling relay, namely, stimulation of both dp53 and JAK/STAT cytokine production. Compared to nontreated $Ras^{V12}$ control tissues, IR increased levels of the dp53 protein in the immunostaining

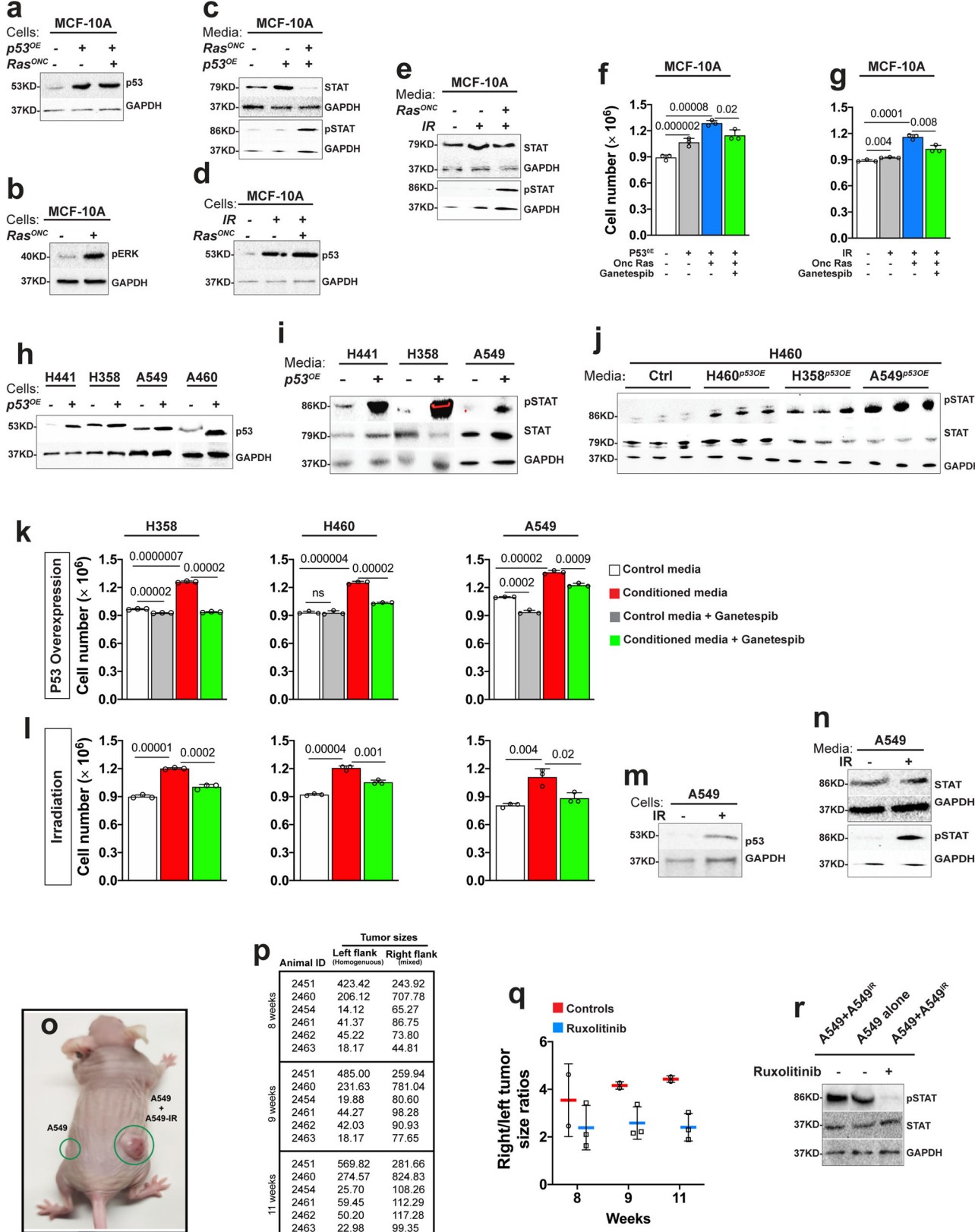

experiments. Notably, dp53 stimulation was nonuniform, and dp53 was undetectable in portions of wild-type (GFP-negative) and $Ras^{V12}$ cells (GFP-positive) (Fig. 5a–b′). This mosaicism supports our finding that $Ras^{V12}$ cells expressing high levels of dp53 protein stimulate the growth of the surrounding $Ras^{V12}$ cells with lower dp53 levels (Fig. 2o-p″). Our qPCR data showed that

IR transcriptionally stimulates all unpaired cytokines ($upd1$-$3$) (Fig. 5e). Similar to $ptip^{-/-}$-induced $upd$ stimulation, IR-triggered upregulation of $upd$ cytokines in $Ras^{V12}$ cells was lost when we introduced dominant-negative dp53 ($p53^{R155H}$) (Fig. 5f). Collectively, our data indicate that irradiation transcriptionally induces the production of $upd$ cytokines downstream of dp53.

**Fig. 4 Wild-type p53 and oncogenic Ras paracrine STAT activation stimulates the growth of human cancer cells.** (**a**–**j**) In vitro experiments showing that media conditioned with *p53*-overexpressing or irradiated cancer cells stimulate cell proliferation via STAT signaling. Western blot of protein lysates prepared from MCF-10A cells expressing oncogenic *HRas* ($Ras^{ONC}$) or wild-type *p53* ($p53^{OE}$) alone or together and blotted with anti-p53 (**a**) or anti-pERK (**b**) or anti-GAPDH (as loading control). (**c**) Western blots showing STAT activity in MCF-10A cells cultured in media conditioned with MCF-10A (controls) or with MCF-10A cells expressing $Ras^{ONC}$ or $p53^{OE}$ alone or together. (**d**) p53 or GAPDH western blot images from MCF-10A cells expressing oncogenic *Ras* or not under normal or irradiated conditions. (**e**) Western blot image showing STAT and GAPDH or phospho-STAT and GAPDH in cells conditioned with media conditioned with irradiated MCF-10A cells or MCF-10A cells expressing $Ras^{ONC}$ from (**d**). (**f**) Conditioned media stimulate cell growth, as determined by cell number under the indicated conditions: MCF-10A cells cultured in media conditioned with MCF-10A control cells or MCF-10A cells overexpressing $p53^{OE}$ or $Ras^{ONC}$ or coexpressing both. Error bars denote SD values. *P* values are derived from Student's *t* test analyses. (**g**) The growth of MCF-10A cells in media conditioned with irradiated MCF-10A cells in the presence or absence of *p53* overexpression is shown. Ganetespib (25 nm) treatment suppressed cell growth. Error bars denote SD values. *P* values are derived from Student's *t* test analyses. (**h**) Western blots showing p53 expression in lung cancer cells transfected with p53 expression construct or not. GAPDH represents the loading control. (**i**, **j**) Western blot images showing STAT activity (pSTAT) in lung cancer cells cultured in media conditioned with unmodified cells or with cells overexpressing *p53*. Total STAT and GAPDH protein levels were used as loading controls. These experiments are shown in biological triplicates in (**j**). (**k**) Effect of media conditioned with *p53* overexpressing lung cancer cells on cell number in the presence or absence of Ganetespib. Ganetespib (25 nm) treatment had no to minimal effect on the growth of control cells but it significantly suppressed the growth of cells growing in conditioned media. (**l**) Effect of media conditioned with irradiated lung cancer cells on cell number in the presence or absence of Ganetespib (25 nm). (**m**) Western blot images showing p53 expression in A549 cells under normal and irradiation conditions. GAPDH represents a loading control. (**n**) Western blot image showing total STAT and pSTAT levels in A549 cells grown in media conditioned with other A549 cells or with irradiated A549 cells. (**o**) Image of a nude mouse showing the size of tumor xenografts (green circles) 8 weeks after flank injection of $1 \times 10^6$ A549 cells. Left flank inoculants consisted of normal A549 cells, while right flank received an equal mixture of normal and irradiated A549 cells. (**p**) Quantification of tumor size from (**o**). Sample size N = 06 animals per group. Tumor sizes (0.523 × length × width × height) were calculated with a digital caliper. To the exception of one animal (animal ID:2451) that developed a larger tumor on the left flank (homogenous inoculants), the remaining five animals developed noticeably larger tumors from the heterogenous inoculants. (**q**) Graphical representation of right to left flank tumor size ratio. At 8 weeks post inoculation, two of the five animals received Ruxolitinib (10 mg/kg) by oral gavage for 3 weeks. The remaining animals were treated with DMSO vehicle control for the same duration. Caliper measurements determined tumor size in treated versus vehicle control animals. Right to left tumor size ratios are shown at 1 and 2 weeks following treatment initiation. (**r**) Western blot from lysates derived from tumor xenografts harvested from animals treated with Ruxolitinib or DMSO vehicle controls were probed with phospho-STAT or STAT or GAPDH antibodies. All error bars denote SD. *P* values are derived from Student's *t* test analyses. Effect size (*d*) values for **f**, **g**, **k** and **l** are greater than 0.8.

We sought to directly test the functional relevance of IR-induced STAT signaling in $Ras^{V12}$ tumor radioresistance in Ras tissues. IR reduced the size of wild-type clones and the overall tissue size, as expected (Fig. 5h, h′, and m). In sharp contrast, IR increased the $Ras^{V12}$ clone size and failed to reduce the overall tissue size (Fig. 5i, i′, m, n, and n′). We extended our analysis to other tumor signaling contexts, tissues containing clones of cells carrying homozygous null mutations in tumor suppressors (*Salvador/Sav* or *tuberous sclerosis/Tsc*) or tissues containing clones of cells overexpressing the oncogene *dMYC*, to determine whether this resistance is unique to oncogenic Ras signaling. In all of these tissues, IR effectively reduced clone and overall tissue sizes (Fig. 5i–l′, and m). Thus, similar to humans, *Drosophila* oncogenic *Ras*-driven tumors are uniquely radioresistant.

Next, we asked to what extent the depletion of dp53 or JAK/STAT cytokines sensitizes $Ras^{V12}$ tissues to IR. We induced $Ras^{V12}$ clones in wild-type or $dp53^{-/-}$ null mutant discs, assessed whether the $dp53^{-/-}$ mutation sensitizes $Ras^{V12}$ tumor tissues to IR and found that it does (Fig. 5n–p, and u).

Next, we simultaneously depleted Upd1 and Upd2 cytokines from $Ras^{V12}$ cells and asked whether this manipulation also sensitizes $Ras^{V12}$ tumor tissues to IR. Two independent approaches were used. First, we introduced an *upd2* null mutation into $Ras^{V12}$ cells coexpressing Upd1-RNAi ($Ras^{V12},upd^{-/-},Upd1$-RNAi clones). Second, we generated clones of cells coexpressing $Ras^{V12}$, *Upd-RNAi* and *Upd2-RNAi*. Both approaches abolished the capacity of $Ras^{V12}$ tissues to grow following IR treatment, supporting a sensitization effect (Fig. 5n, n′, q–r′, and u). Similarly, specific inhibition of the JAK-STAT receptor *domeless* in $Ras^{V12}$ cells via the expression of a potent dominant-negative protein version (*domeless-DN*)[62] sensitized $Ras^{V12}$ tissues to IR (Fig. 5n, n′, s–t′, and u).

Thus, in *Drosophila* and human cancer cells, the p53 response to genomic instability cooperates with oncogenic Ras to induce JAK/STAT activation in surrounding cells. This nonautonomous effect stimulates tumor growth and promotes the rapid recurrence of oncogenic Ras tumors.

## Discussion

Oncogenic Ras mutations are associated with resistance to genotoxic therapies. The underlying resistance mechanism remains poorly understood. The molecular responses of tumor cells to genotoxic stress have been investigated mainly in isolated cells, which limits our ability to capture the broader tissue-level tumor biology.

Using *Drosophila* eye tissue in a clonal genetic screen for $Ras^{V12}$ suppressors, we unexpectedly isolated the genotoxic mutation $ptip^{-/-}$. Interestingly, in addition to blocking Ras tumor growth, $ptip^{-/-}$ stimulates the surrounding $Ras^{V12}$ tissue to overgrow, mirroring the resistance of $Ras^{V12}$ tumors to genotoxic therapies. This nonautonomous effect stems from cooperation between oncogenic Ras signaling and $ptip^{-/-}$, as oncogenic Ras or $ptip^{-/-}$ mutant cells alone do not cause non-autonomous growth. Our mechanistic studies reveal that this cooperation is centered on p53.

P53 is broadly known as a tumor suppressor gene. The majority of *p53* mutations in human cancers are missense mutations that stabilize the p53 protein, leading to elevated levels of the mutant p53 protein in cancers. These gain-of-function mutations interfere with the canonical tumor suppressor role of p53 while causing it to function as an oncogene[63–66]. The accumulation of mutant p53 is associated with aggressive cancers[63–67]. Interestingly, overexpression of wild-type p53 is also observed in many cancers lacking *p53* mutations, including in lung cancers where oncogenic Ras mutations are common[68–72]. How wild-type p53 overexpression relates to oncogenic Ras cancers and their resistance to genotoxic therapies remained unclear. Here, we show that genotoxic stress-activated p53 acts non-cell autonomously to promote the radioresistance of Ras mutant tumor tissues.

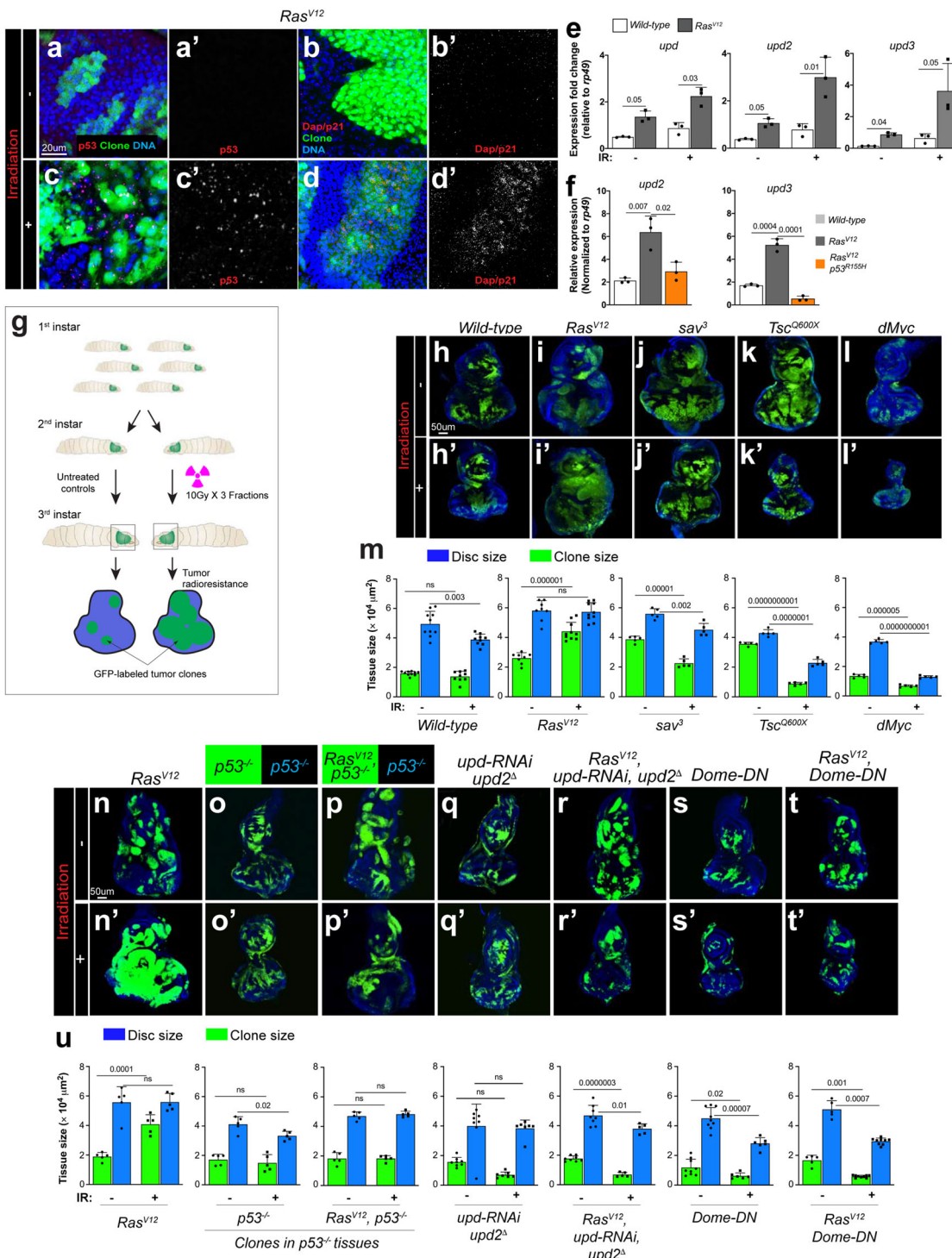

The $ptip^{-/-}$ mutation causes genomic instability in $Ras^{V12}$ cells, resulting in the upregulation of the dp53 protein. This stimulation of p53 is essential for the nonautonomous tissue overgrowth effect of $Ras^{V12}ptip^{-/-}$ tumor clones. RNAi depletion of dp53 in $Ras^{V12}ptip^{-/-}$ clones abrogates the nonautonomous tissue overgrowth effect. In addition, direct overexpression of dp53 in $Ras^{V12}$ clones is sufficient to trigger overgrowth of the surrounding tissues, mimicking the $ptip^{-/-}$ mutation.

The nonautonomous tissue overgrowth effect is mediated by the dp53 transcriptional program. Expression of a DNA binding-defective dp53 mutant ($p53^{R155H}$)[53] in $Ras^{V12}ptip^{-/-}$ cells blocks

the nonautonomous tissue overgrowth effect. Consistent with this finding, the transcriptional program of wild-type p53 is modified in cancer-associated fibroblasts to promote cancer progression[73]. Additionally, under ectopic wild-type p53 conditions, oncogenic Ras modifies the p53 transcriptional program, leading to the senescence-associated secretory phenotype (SASP)[74–76].

JAK/STAT cytokines are the primary transcriptional targets relevant to the cooperative effect of Ras/p53 on nonautonomous growth. JAK/STAT cytokines are transcriptionally upregulated in tissues showing nonautonomous growth ($Ras^{V12}ptip^{-/-}$ or $Ras^{V12}p53^{OE}$) compared to controls ($Ras^{V12}$). The specific

**Fig. 5 The Wild-type p53/oncogenic Ras nonautonomous STAT signal relay promotes the radioresistance of *Drosophila* Ras tumor tissues. (a–d′)** Upregulated p53 and dap/p21 within *Ras^V12* clones after irradiation. GFP-labelled *Ras^V12* clones were stained with anti-p53 antibody (**a**, **a′**, **c**, **c′**) or anti-p21 antibody (**b**, **b′**, **d**, **d′**) before irradiation (0 h) and 24 h after irradiation. Time was counted from the start of first faction of IR treatment. Scale bar is 20 μm. (**e**, **f**) Quantification by qPCR of *upd*, *upd2*, and *upd3* expression in eye-antennal discs containing wild-type or *Ras^V12* clones after 36 h of first fraction of IR treatment (IR+) or without IR treatment (IR−). Column bars represent the mean of fold changes for the expression level of indicated genes (**e**). Relative expression of *upd2* and *upd3* in irradiated eye-antennal discs containing wild type, *Ras^V12* and *Ras^V12p53^R155H* clones (**f**). Three independent experiments were carried out. Error bars denote SD. *P* values are derived from Student's *t* test analyses. (**g**) Diagram of setting *Drosophila* irradiation models. Larvae after egg laying (48 h) were irradiated with three fractions of 10 Gy and allowed to recover to late third-instar larval stage. All eye-antennal discs were dissected at the late third-instar larval stage to evaluate the irradiation results by measuring the relative size between GFP-labeled clones and whole eye-antennal discs. (**h–l′**) GFP-labeled clones homozygous for *Ras^V12* (**i**, **i′**), *sav^3* (**j**, **j′**), *Tsc1^Q600X* (**k**, **k′**), or expressing *dMyc* (**l**, **l′**) as well as wild-type controls (**h**, **h′**) were induced in the eye-antennal discs of larvae, irradiated at 48 h, and then collected discs on day 5. (**h–l**) the eye-antennal discs without irradiation treatment (IR−). (**h′–l′**) show irradiated discs (IR+). (**m**) Quantification of relative eye disc size (blue) and GFP-clone size (green) treated with IR (IR+) or without IR (IR−). For each genotype, eye-antennal disc and GFP-clone were normalized to age-matched eye discs without IR. Column bars represent the mean size of samples (N = 5–10). Scale bar is 50 μm. (**n–t′**) GFP-labeled *Ras^V12*, *p53^−/−*, *Ras^V12p53^−/−*, *upd^RNAi upd2^Δ*, *Ras^V12Upd^RNAi upd2^Δ*, *Dome^DN*, and *Ras^V12Dome^DN* clones were induced in the eye-antennal discs and half were then irradiated at the second-instar larval stage. After 3 days of recovery, all eye discs at the late third-instar larval stage were dissected to evaluate the differences in response to irradiation. (**n–t**) Eye-antennal discs without irradiation treatment (IR−) and (**n′–t′**) eye discs treated with irradiation (IR+). Scale bar is 50 μm. (**u**) Quantification of clones and eye discs treated with or without irradiation. Eye-antennal disc and GFP-clone areas were measured by ImageJ and normalized to the eye-antennal discs with the same genotype at the same age without IR. Column bars represent the mean size of samples (N = 5–9). Blue columns represent the mean size of the entire eye-antennal tissue for the indicated genotypes; green columns represent the size of GFP-labeled tumors. Error bars denote SEM. *P* values are derived from Student's *t* test analyses. Effect size (*d*) values for **e**, **f**, **m**, and **u** are greater than 0.8.

depletion of Upd cytokines in *Ras^V12ptip^−/−* or *Ras^V12p53^OE* clones suppresses nonautonomous tissue overgrowth. Increases in wild-type p53 levels either via IR, the *ptip^−/−* mutation alone, or direct *dp53* overexpression are sufficient to stimulate *upd* cytokine production (Fig. 5e and Supplementary Fig. 5c–e), but these effects do not cause nonautonomous growth, likely because these cells are quickly eliminated in the absence of oncogenic Ras.

The cooperative effect of oncogenic Ras and p53 on paracrine STAT signaling may also reflect an ability of Ras to rewire the dp53 transcriptional program and/or to increase the exocytosis of STAT cytokines above a required threshold. Consistent with these possibilities, oncogenic Ras stimulates exocyst in flies and mammals, and we identified and validated dp53-binding sites near *upd* genes. Interestingly, the deletion of these sites significantly suppressed *upd* expression but did not restore *upd* expression to basal levels in reporter assays (Supplementary Fig. 5), possibly because of cryptic dp53-binding sites located near *upd* genes. Notably, p53 binds to noncanonical DNA sites to expand its transcriptional network[77]. In the future, it would be desirable to map these noncanonical p53 sites on *upd* in order to better understand p53 function.

The nonautonomous Ras/p53-STAT signaling relay allows Ras mutant clones to resist the damaging effects of IR treatment in *Drosophila*, as dp53 or STAT depletion sensitizes Ras mutant tumor tissues to IR. Wild-type dp53 is stimulated nonuniformly in irradiated Ras mutant tissues. This heterogeneity might be due to stochastic variation or reflect different cell-inherent capacities to successfully withstand genotoxic stress. Treatment-induced p53 heterogeneity within Ras mutant tumor tissues would allow cells with extensive genomic insults (high dp53 levels) to directly induce the upregulation of JAK/STAT ligands, which instructs nearby less-damaged (low dp53) *Ras^V12* cells to overproliferate and reestablish the tumor following treatment.

In vitro supernatant transfer and mouse xenograft experiments revealed a similar mechanism in human Ras cancer cells. Indeed, activation of STAT signaling is associated with resistance to genotoxic agents in human cancer cells[78–80]. Compared to controls, media conditioned with irradiated or p53-overexpressing Ras cancer cells elevate STAT signaling and stimulate cell proliferation across genetically diverse cancer cells. We propose that the Ras-p53 nonautonomous STAT signaling relay likely represents a tissue-level adaptive mechanism for selecting and

expanding therapy-resistant tumor clones in the tumor microenvironment. This mechanism is reminiscent of the paracrine activation of TGFα/amphiregulin signaling by oncogenic Ras to establish resistance to EGFR blockade in colorectal cancers[81].

In addition to highlighting an emerging role for p53 in cell–cell interactions, our findings provide a possible explanation for the paradoxical resistance of *Ras* cancers to genotoxic therapies, despite functional p53[82,83]. Thus, our data suggest that combining STAT inhibition with radiation therapy may improve clinical outcomes.

Developmental and regenerative signaling contexts may functionalize p53 in a similar manner to maintain tissue homeostasis. Neighboring cells with different levels of wild-type p53 influence the growth of other cells in *Drosophila* and mammalian tissues[84].

## Methods

**Fly strains and generation of clones.** Flies were raised on standard *Drosophila* media at 25 °C. Fluorescently labelled mitotic clones were produced in larval imaginal discs using the following strains: (1) *yw, eyFLP1; Act>y + >Gal4, UAS-GFP.S65T; FRT82B, Tub-Gal80*; (2) *yw; eyFLP5, Act>y + >Gal4, UAS-GFP; FRT82B, Tub-Gal80*; (3) *FRT42D, Tub-Gal80*; (4) *yw, upd2^Δ3-62;eyFLP5, Act>y + >Gal4, UAS-GFP; FRT82B, Tub-Gal80*; (5) *yw, eyFLP1; Act>y + >Gal4, UAS-GFP.S65T; FRT79E*; (6) *yw, eyFLP1; Act>y + >Gal4, UAS-GFP.S65T; Tub-Gal80, FRT79E*; (7) *yw, eyFLP1; Act>y + >Gal4, UAS-GFP.S65T; Tub-Gal80, ptip^3804, FRT79E* and (8) *yw,eyFLP1;Act>y + >Gal4,UAS-GFP;Tub-Gal80,FRT79E*. Additional strains used were as follows: (1) *yw; FRT82B*; (2) *w;UAS-Ras^V12(II)*; (3) *w; UAS-Ras^V12(III)*; (4) *w;; FRT82B,Tsc1^Q600X/TM6B*; (5) *yw; 82B, sav^3/TM3*; (6) *yw; UAS-dMyc; Sb/TM6B*; (7) *UAS-dp53/CyO* (gift from N. Senoo-Matsuda); (8) *yw; UAS-p53^R155H/T(2;3)TSTL, CyO: TM6B, Tb* (Bloomington Stock Center, BL8419); (9) *yw; p53 ^5A-1-4* (BL6815); (10) *UAS-p53-RNAi* (VDRC, v103001); (11) *yw;FRT42D,ubi-RFP.nls*; (12) *w,UAS-Bsk^DN*; (13) *yw; UAS-Ras^V12, FRT79E,ptip^3804*; (14) *w; UAS-Upd-RNAi(R1)* (III) (NIG5988R); (15) *yw,upd2^Δ3-62* (gift from M. Zeidler); (16) *yw,upd-lacZ* (gift from H. Sun); and (17) *w,UAS-dome^Δcyt1.1* (gift from J. Hombria).

***Drosophila* x-ray irradiation.** Second-instar larvae (54 ± 6 h after egg laying) were collected from a petri dish containing 3 ml food and irradiated with three fractions of 1000 R at 6 h intervals in a Cabinet Faxitron X-Ray machine (TRX 2800). All larvae were allowed to recover at 25 °C. Eye discs were dissected at the late third-instar stage (day 5) based on developmental markers (spiracles and mouth hooks). For timing experiments, second-instar larvae were irradiated using three fractions of 1000 R (10 Gy), and dissections were timed from the end of treatment.

**Staining and imaging.** Antibody staining was performed according to standard procedures for *Drosophila* imaginal discs. The following antibodies were used: mouse monoclonal anti-β gal (1:500, Sigma), mouse anti-dmp53 (1:50 DSHB), rabbit anti-Stat92E (1:1000; gift from S. Hou), rabbit anti p-JNK (1:100), mouse

anti-p21 (1:200, DSHB) and mouse anti-H2Av monoclonal antibody (1:200, DSHB #UNC93-5.2.1). Secondary antibodies were purchased from Life Technologies. Images were acquired on a Leica SP8 confocal microscope. Measurements of tumor clone size within imaginal discs were performed from confocal pictures using Fiji imageJ software. Adult eyes were imaged with a Leica DFC 300FX camera in a Leica MZ FLIII fluorescence stereomicroscope.

**Real-time RT-PCR**. Total RNA from wild-type and tumor imaginal discs ($n \geq 30$ pairs) were extracted using Trizol. cDNAs were synthesized from 500 ng total RNA with the iScript cDNA Synthesis Kit (Bio-Rad). cDNAs were subjected to real-time PCR with the SYBR Green Fast Kit (Applied Biosystems), according to the manufacturer's instructions. The expression level of genes for each sample was calculated by comparing to the internal control, *rp49*. The relative fold change of each gene was normalized to the expression level of the same gene in the eye disc bearing *Ras^V12*. Three experiments for each condition were averaged. The following primers were used for qRT-PCR:

*upd*: 5'-TCCACACGCACAACTACAAGTTC-3'; 5'-CCAGCGCTTTAGGGCA ATC-3'

*upd2*: 5'-AGTGCGGTGAAGCTAAAGACTTG-3'; 5'-GCCCGTCCCAGATAT GAGAA-3'

*upd3*: 5'-TGCCCCGTCTGAATCTCACT-3'; 5'-GTGAAGGCGCCCACGTAA-3'

*dp53*: 5'-CTATTGAGCTGGCGTTCGTCTTGGAT-3'; 5'-TCTGCCAAAAC TCGTGTATCGGGCG-3'

*dap/p21*: 5'-GTCAGCTTCCAGGAGTCGAG-3'; 5'-CCAAAGTTCTCCCGTT CTGA-3'

*rp49*: 5'-GGCCCAAGATCGTGAAGAAG-3'; 5'-ATTTGTGCGACAGCTTA GCATATC-3'

**Luciferase assay**. The XmaI-XbaI fragment, including the predicted p53 binding sites, from upstream of Upd2 was separated from BAC DNA (BACR32F24) by enzyme digestion and then ligated into the pGL3-promotor to get the pGL3-Upd2-luc reporter construct (Upd2-p53BDS). The upstream fragment of Upd3 was separated by XmaI-SpeI enzyme digestion from BAC DNA and then ligated into the pGL3-promoter plasmid to form the pGL3-upd3-luc (Upd3-p53BDS). The predicted p53 binding sites were deleted from the reporter constructs to generate pGL3-upd2$^\Delta$-luc (Upd2-p53BDS$^\Delta$) and pGL3-upd3$^\Delta$-luc (Upd3-p53BDS$^\Delta$). $1.6 \times 10^5$ Schneider S2R + cells per well were plated in a 24-well plate one day before transfection. S2R + cells were transfected with pGL3-Upd2-luc and pGL3-Upd3-luc reporter constructs and the p53 binding site deletion constructs separately along with metallothionein promoter-dp53 constructs (MT-p53) (from DGRC, FMO05476) and Renilla reporter (pRL-TK Vector) by using the X-treme GENE HP DNA Transfection Reagent (Roche). After 48 h of transfections, copper sulphate was added to the medium in a final concentration 500 µM to induce dp53 expression. After the inducing expression of dp53 for 24 h, a dual luciferase assay was performed. Relative expression levels were calculated based on the Renilla reporter as an internal control. Experiments were carried out using three or more biological replicates.

**Cell lines and cell culture**. H441, H358, H460, and A549 were authenticated at the Cell and Immunology Core facility at the University of Missouri. MCF-10A and MCF-10A^Onc HRAS cells were a gift from G. Monogarov (DKFZ, Heidelberg, Germany). Cells were cultured following ATCC recommendation in DMEM media containing high glucose and L-glutamine (Gibco#11875-093) or in RPMI-1640 media for H441 cells. A549 cells were cultured in Dulbecco's Modified Eagle Medium with L-glutamine and high glucose (DMEM#11965-092) supplemented with 10% FBS. MCF-10A cells were cultured in DMEM-F12 (11320-033) with 5% horse serum (Gibco, 16050-122), hydrocortisone 0.5 µg/ml, cholera toxin 100 ng/ml, insulin 10 µg/ml, and EGF 20 ng/ml. Cells were incubated in a humidified incubator with 5% $CO_2$ at 37 °C.

**Cell counting**. TrypLE (Gibco#12604-021) was used for detaching cells. Cells were counted using Trypan Blue as a dead cell excluder (Cat #T8154, Sigma USA).

**Transfection**. The p53-GFP vector was purchased from addgene (#1209). DharmaFECT (Dharmacon# T2002-02) was used for transfecting the p53-GFP vector into the human cell lines, following the manufacturer's recommendation.

**Ganetespib cytotoxicity**. MTT assays on cancer cells treated with increasing concentrations of Ganetespib (0.1–100 nM) determined the half maximal inhibitory concentration (IC50) at 25 nM. Cells were seeded in 96-well plates at a density of $1c10^5$ cells per well and treated with Ganetespib (Biosciences, #A11402,) for 48 h. A 100 µl solution of MTT (100 µg/ml) was added to each well. Cell viability was measured with a spectrophotometer at 570 nm.

**Immunoblotting**. For the immunoblotting analysis, cells were lysed in a buffer (20 mM Tris-HCl pH-7.5, 150 mM NaCl, 1 mM Na$_2$EDTA, 1 mM EGTA, 1% Triton, 2.5 mM sodium pyrophosphate, 1 mM b-glycerophosphate, 1 mM Na3VO4, 1 µg/ml leupeptin) supplemented with protease and a phosphatase inhibitor cocktail

(Cell Signaling #9803S). Absolute protein concentration was determined and normalized using a BSA standard curve on Nanodrop (Thermo Fisher Scientific). Proteins were electrophoresed on SDS-PAGE using 4–15% Mini-PROTEAN® TGX™ precast gel (Biorad#456-1085) and transferred onto a nitrocellulose membrane using Trans-Blot® Turbo™ Transfer Pack (Biorad#170-4159). The nitrocellulose membrane was blocked in buffer (5% milk in Tris-buffered saline containing 0.01% Tween20). Membranes were probed overnight at 4 °C with anti-pSTAT-3 (Tyr 705) (1:1000, Cell Signaling #9145) or anti-STAT (1:1000, Cell Signaling #30835) or anti-GAPDH (1:1000, Cell signaling #5174S) or anti-pERK1/2 (12D4, 1:1000, SantaCruz #81492). Secondary horseradish peroxidase (HRP) antibodies were obtained from Invitrogen. An ECL Chemiluminescence Kit (Thermo Fisher Scientific #32106) and the ChemiDoc Imaging System (Bio-Rad) were used to detect protein bands.

**Human cancer cells irradiation**. For x-irradiation conditioned media experiments, cells were cultured in fresh media prior to irradiation (8 Gy, 280 cGy/min exposure) using an X-RAD 320 Biological Irradiator. Media were replaced immediately following irradiation, and the irradiated cells were cultured for 24 h to generate conditioned media. Supernatant from the irradiated cells was collected after centrifugation at $500 \times g$ for 2 min to remove cellular debris.

**Flow cytometry analysis**. GFP cell sorting was performed on the MoFloXDP (Beckman Coulter), and the regular flow cytometer analysis was performed using a CyAn ADP (Beckman Coulter). Flow cytometry cell proliferation assays were performed using the 5-ethynyl-2'deoxyuridine (EdU) assay kit (Invitrogen #C10634). Cells were incubated with EdU for 24 h and labeled. The Click-iT reaction was performed using the Click-iT EdU assay kit with Alexa Fluor 647 fluorophore, according to the manufacturer's instructions. The propidium iodide (PI) solution was at 10 µg/ml in PBS containing 1% BSA. Gating was set at 488/636 nm (excitation/emission) or at 633/660 nm to detect and score PI or EdU_Alexa647-positive cells, respectively.

**Mouse xenograft experiments**. The mouse study was performed with the approval and oversight of the Institutional Animal Care and Use Committee at the University of Missouri (Protocol#9501). 13-weeks old Fox n1<Nu> homozygous male and female mice were used. Thirteen-week-old athymic nude mice (homozygous $Foxn1^{nu}$) were purchased from the Jackson Laboratory for xenografts experiments. Experiments were conducted in compliance with the National Institute of Health's guide for the care and use of animals. All animals were housed under pathogen-free conditions on a 12/12 h light-dark cycle. The A549 cells were grown in Dulbecco's Modified Eagle Medium with L-glutamine and high glucose supplemented with 10% FBS. Cells were collected in pharmaceutical grade PBS, counted, and resuspended in pharmaceutical grade PBS at $1 \times 10^6$/100 µL. All the left flanks were inoculated with $1 \times 10^6$ A549 cells. Right flanks were inoculated with a 50/50 mixture of A549 and irradiated A549 ($0.5 \times 10^6 + 0.5 \times 10^6$ cells) in a total of 100 µL. Tumor measurements were taken weekly using a digital caliper. The STAT inhibitor Ruxolitinib (Sigma Millipore #ADV390218177) was administered (10 mg/kg body weight) through oral gavage once a day for a period of three weeks.

**Statistics and reproducibility**. All experiments were performed in three biological replicates for reproducibility. Standard deviations represent at least three biological replicates. Student's *t* test was used to determine the statistical significance of differences between groups. Effect size was determined by calculating Cohen's *d* value [$d$ = |mean$^{(group1)}$-mean$^{(group2)}$|/Pooled standard deviation].

## Data availability

All source data are provided in Supplementary Data 1. All other data are available from the corresponding author on reasonable request.

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

## Acknowledgements
The authors thank M. Oren, G. Monogarov, J. Contessa, B. Dunn, S. Ding, F. Qian, H. Chang, and D. Manry for helpful discussions. We are grateful to M. Oren and G. Monogarov for providing MCF-10A cells. We also thank N. Senoo-Matsuda, H. Zeidler, J. Sun, S. Hombria, S. Hou, the Bloomington Stock Center, the Vienna *Drosophila* RNAi Center, and the National Institute of Genetics for fly strains and antibodies. T.K. was supported by an RSNA research grant. This work was supported in part by the Chinese National Natural Science Funds for Young Scholars (No. 31200687) to Y.D., the leading Innovative and Entrepreneur Team Introduction Program of Zhejiang (2018R01003) and Westlake University (TX). T.X. was an investigator at the Howard Hughes Medical Institute. The work was supported by start-up funds to CYC from the University of Missouri.

## Author contributions
Y.D., T.K., J.L., P.G., T.X., and C.Y.C. conceived the study. Y.D. and T.K. developed the fly irradiation models. L.L. assisted with qRT-PCR and fly stock maintenance. Y.D. and V.A. performed fly experiments. G.P. performed the tissue culture and mouse xenografts experiments. C.Y.C. wrote the manuscript.

## Competing interests
The authors declare no competing interests.
