## [Peer Review File · Communications Biology]

Reviewers' comments:

Reviewer #1 (Remarks to the Author):

The manuscript by Dong et al. depicts the identification of a non-cell autonomous mechanism that promotes activation of JAK/STAT signaling via wild-type p53 and activated RAS. The authors first identified *ptip* as a gene whose absence blocked tissue overgrowth induced by oncogenic RAS in the developing *Drosophila* eye epithelium. Loss of *ptip* induced DNA damage in RAS-mutant cells which resulted in p53 stabilization and activation of STAT signaling via upregulation of cytokines that promote JAK/STAT activation. Genetic experiments show that p53 transcriptional activation is the crucial step in this process and that other sources of p53 activation, such as ionizing radiation, cause the same effect. Hence, they propose that this mechanism could contribute to increased resistance to radiotherapy induced by RAS oncogenes.

Overall, the manuscript describes an interesting scenario that underlies a novel non-cell autonomous mechanism that may be involved in promoting radioresistance in tumor cells carrying wild-type p53. Indeed, the results even may provide novel options to treat radioresistant tumors, i.e. via inhibition of STAT signaling. Whereas the data describing this mechanism in flies is mostly convincing, its description in human cancer requires somewhat more solid data.

Furthermore, the manuscript lacks several controls as well as additional insights until it can be published:

1. In Figure 2b, the authors use γ H2AX immunostaining to detect DNA damage. However, it is actually not clear whether absence of *ptip* really induces DNA damage or some other condition detected by γ H2AV immunostaining. Does lack of *ptip* indeed provoke DNA damage? Moreover, does this also occur in the absence of RAS? If so, does it induce p53 under this condition?

2. In mice, *Ptip* was shown to act as a tumor suppressor (Wu et al., PNAS 155:E3978-86; 2018). However, it appeared that this activity is connected with the epigenetic regulator UTX. Therefore, it cannot be ruled out that in RAS-mutant cells in *Drosophila*, *ptip* also acts as a tumor suppressor and actually cooperates with RAS to induce γ H2AV and p53 staining. In other words, is it possible that in RASV12, *ptip* KO cells some kind of senescence response is observed? If not, how can this be ruled out? Also in relation to point 1, is this response really related to DNA breaks or could it be some kind of replicative stress?

3. In Figure 4, a series of controls are lacking: In Figs. 4a and 4b, the levels of total STAT3 are missing. In addition, it would be relevant to show what are the levels of p53 and RAS in the cells used to obtain conditioned media. Fig. 4F lacks GAPDH and Fig. 4i total STAT3. In Fig. 4k, it would be better to demonstrate the absolute numbers of both left and right tumors, to appreciate whether the drug also inhibited the non-irradiated tumors. Moreover, the activation status of STAT3 in each of these tumors before and after treatment with Ruxolitinib should be demonstrated.

4. In a mouse model of pancreatic cancer, it was shown that absence of p53 rather than p53 stabilization promotes STAT3 activation in combination with RAS via accumulation of ROS (Wörmann et al., Gastroenterology 151:180-93; 2016). What is the alternative mechanism of non-cell autonomous STAT3 activation upon p53 stabilization in human cells in Figure 4? Which cytokines may be involved?

5. According to the data presented in this manuscript, p53 only promotes STAT3 activation in the context of oncogenic RAS. What is the precise contribution of RAS signaling? How does it modify the

p53 transcriptional program? Is this effect determined by some threshold of p53 activation? Could other factors be involved?

6. In Figure 5C it appears that IR by itself is sufficient to induce the unpaired cytokines, and that RAS activation stimulated his effect even further. Is the same observed upon p53 OE (Fig. 3d) or ptp absence (Fig. 3e) alone?

7. I would propose to change the title to: "Cooperation between oncogenic RAS and wild-type p53 stimulates JAK/STAT non-cell autonomously to promote tumor radioresistance.", because it could be relevant to mention that wild-type p53 rather than mutant p53 plays a role and "RAS tumor radioresistance" is not a well-defined term. However, this is only a suggestion and may not be followed if the authors don't agree.

Reviewer #2 (Remarks to the Author):

Chabu, Xu and colleagues use a *Drosophila* epithelial model (the eye imaginal disc) to carry out an enhancer/suppressor genetic screening of RasV12-driven tumorigenesis and present evidence that mutations in *ptip1*, which are poorly recovered under normal circumstances, promote non-autonomous tissue growth when combined with RasV12-overexpression. Authors present evidence that these clones have high levels of Dp53 and ectopic expression of Upd cytokines and demonstrate that the Dp53-Upd axis has a major role in RasV12/*ptip*-driven non-autonomous overgrowth. In this context, it would be necessary to see whether Dp53 is transcriptionally activated (or just the protein stabilized) and whether bona fide transcriptional reporters of Dp53 are being induced. Also, epistatic relationship to show that Upd expression relies on the activity of Dp53 in RasV12/*ptip* mutant clones should be shown. Similarly, a role of Upds in RasV12/*ptip*-driven overgrowth should be experimentally validated. Authors validate their observations in cancer cell lines and xenograft experiments and, in the last part of the ms, authors present evidence that ionizing radiation (known to cause DNA damage) cooperates with RasV12 to promote tissue growth. Nicely, authors demonstrate that Dp53 protein and Upd expression levels are also increased in IR-treated RasV12 tissues and that Dp53 and Upds are required for the growth of RasV12 expressing cells upon IR treatment. The ms is well written, figures are self-explanatory, and experimental data require some epistatic analysis to demonstrate their claim not only in the RasV12/*ptip* condition but also in the RasV12/IR experimental setting. In this regard, experiments concerning the regulation of Upds by Dp53 downstream of Ras/IR and bona fide reporters of Dp53 should be added. Unfortunately, the major message of the ms - as stated in the title, abstract, results and discussion on the role of Dp53 and STAT depletion in radiosensitizing RAS tumors - is not substantiated by the data. Authors present evidence that Dp53 and STAT are required for the growth of IR-treated RAS clones (most probably as a consequence of the well-known mitogenic activity of Upds). Authors should present data on cell death to demonstrate that depletion of Dp53 and STAT sensitizes RAS-clones to IR-induced cell death.

Reviewer #3 (Remarks to the Author):

The authors of the manuscript posit a connection between genotoxic stress and oncogenic RAS activation leading to cell proliferation in the tumor microenvironment via a p53 mediated JAK/STAT activation pathway. Interestingly, they argue that these results can explain the paradoxical tumor radioresistance in Ras tumours with wt p53, and that a possible avenue of therapy rescue may be co-treatment with JAK/STAT inhibitors. The authors identify this mechanism in an animal model using *Drosophila* genetic screening, where a uniquely genotoxic mutation *ptip*^{-/-} acts to destabilize the

genome, but also promote overgrowth of surrounding wildtype tissue via p53. The approach is interesting, as using animal models in this way may reveal complex cell-cell interactions dependant on a complete tissue microenvironment otherwise lacking in normal cell cultures. The authors further confirm this finding by specific gene regulation studies using qPCR, as well as human breast and lung cancer monolayer cell cultures and in vivo mouse xenograft models together with ionizing radiation for genotoxicity. Importantly, the effect persisted in all model systems across species, indicating a fundamental and evolutionary preserved mechanism of action. Most significantly, existing drugs targeting STAT signaling could inhibit the nonautonomous growth-inducing effect of p53 and ionizing radiation in cell cultures (Ganetespib) and mice (Ruxolitinib). The authors lastly closely examine the cellular signaling pathways and elucidate the uniqueness of the described interaction, in comparison to e.g. Salvador/Sav etc.

All in all, the presented work is a thorough and convincing body of evidence of the connection between oncogenic Ras mutations and cell proliferation in combination with genotoxic treatments such as ionizing radiation, mediated by p53 and JAK/STAT induction.

- * The paper is technically sound
- * The claims are convincing
- * The claims are on the whole supported by the experimental data presented
- * The statistical analysis of the data is sound
- * The claims are appropriately discussed in the context of previous literature

Minor concerns:

As the authors know, the investigated p53 levels are finely regulated by members of the MDM- family, like MDM2 and MDMX. What are the levels of these proteins and does their expression affect radio-sensitivity and apoptotic signalling? At what point do proteins of the negative feedback loop peak and what role do they play in p53 regulation in the model presented?

Response to reviewers' comments

Manuscript#: COMMSBIO-20-0984-T

Title: Cooperation between oncogenic Ras and p53 stimulates JAK/STAT non-cell autonomously to promote Ras tumor radioresistance"

Reviewer #1 (Remarks to the Author): The manuscript by Dong et al. depicts the identification of a non-cell autonomous mechanism that promotes activation of JAK/STAT signaling via wild-type p53 and activated RAS. The authors first identified *ptip* as a gene whose absence blocked tissue overgrowth induced by oncogenic RAS in the developing *Drosophila* eye epithelium. Loss of *ptip* induced DNA damage in RAS-mutant cells which resulted in p53 stabilization and activation of STAT signaling via upregulation of cytokines that promote JAK/STAT activation. Genetic experiments show that p53 transcriptional activation is the crucial step in this process and that other sources of p53 activation, such as ionizing radiation, cause the same effect. Hence, they propose that this mechanism could contribute to increased resistance to radiotherapy induced by RAS oncogenes.

Overall, the manuscript describes an interesting scenario that underlies a novel non-cell autonomous mechanism that may be involved in promoting radioresistance in tumor cells carrying wild-type p53. Indeed, the results even may provide novel options to treat radioresistant tumors, i.e. via inhibition of STAT signaling. Whereas the data describing this mechanism in flies is mostly convincing, its description in human cancer requires somewhat more solid data. Furthermore, the manuscript lacks several controls as well as additional insights until it can be published:

1. In Figure 2b, the authors use γ H2AX immunostaining to detect DNA damage. However, it is actually not clear whether absence of *ptip* really induces DNA damage or some other condition detected by γ H2AV immunostaining. Does lack of *ptip* indeed provoke DNA damage? Moreover, does this also occur in the absence of RAS? If so, does it induce p53 under this condition?

Phosphorylation of a Histone 2A variant (γ H2Av) is a reliable and widely used indicator of DNA damage. Following the reviewer's request, we performed immunostaining experiments using anti phospho- γ H2Av antibodies on *ptip* mosaic tissues and detected elevated levels of phospho- γ H2Av the mutant clones compared to surrounding tissues. This is consistent with *Ptip* known role in maintaining genomic integrity in mammalian cells.

As requested by the reviewer, we examined the effect of *ptip* mutation on p53 induction and found that *ptip* mutant tissues transcriptionally upregulate p53 in qPCR experiments, congruent with DNA damage in these cells and *PTIP* known role in genome stability maintenance in mammalian cells¹⁻³. These findings and the corresponding figures have been added to the manuscript (Page 6, lines 123-127).

2. In mice, *Ptip* was shown to act as a tumor suppressor (Wu et al., PNAS 155:E3978-86; 2018). However, it appeared that this activity is connected with the epigenetic regulator UTX. Therefore, it cannot be ruled out that in RAS-mutant cells in *Drosophila*, *ptip* also acts as a tumor suppressor and actually cooperates with RAS to induce γ H2AV and p53 staining. In other words, is it possible that in RASV12, *ptip* KO cells some kind of senescence response is

observed? If not, how can this be ruled out? Also in relation to point 1, is this response really related to DNA breaks or could it be some kind of replicative stress?

Knocking down Ptip specifically in oncogenic Ras (Ras^{V12})-expressing cells suppresses Ras^{V12} -mediated tumor overgrowth in *Drosophila* (Fig1a-d, GFP-labelled clones), indicating that Ptip does not act as a cell-autonomous tumor suppressor in *Drosophila*. However, *ptip* acts as a non-cell autonomous tumor suppressor in the context of Ras^{V12} clones. Knocking down *ptip* in Ras^{V12} cells triggers the overgrowth of the surrounding cells (Figure 1e,f, k, l, p-s). In contrast to UTX, which Ji and colleagues showed acts as a cell autonomous tumor suppressor in a mouse lung cancer model⁴, the precise function of Ptip remains unknown. Our data suggests that the increased Ras^{V12} tumor burden caused by Ptip depletion is likely a result of the *ptip* knockdown condition driving the growth of the surrounding Ras^{V12} cells. Indeed, juxtaposition of Ras^{V12} cells with other Ras^{V12} cells with Ptip knocked down leads to a significant overgrowth of the Ras^{V12} tissues (Supplementary Figure 1a-d).

We performed senescence associated- β -galactosidase staining experiments to address the possibility that the *ptip* mutation triggers senescence and it doesn't appear to be case. Consistent with oncogene-induced senescence and previous reports^{5,6}, we detected significant signals in Ras^{V12} clones. The *ptip* mutation did not elevate the β -gal signals in Ras^{V12} clones (See figure right below).

We believe that γ H2Av positivity primarily reflects DNA damage rather than *some other conditions or replicative stress*. In the MARCM clones eyFlp introduces the *ptip* mutation very early in eye development, severely limiting the ability of Ras^{V12} cells to over proliferate (Figure 1a-d in the manuscript) and induce any significant replicative stress.

3. In Figure 4, a series of controls are lacking: In Figs. 4a and 4b, the levels of total STAT3 are missing. In addition, it would be relevant to show what are the levels of p53 and RAS in the cells used to obtain conditioned media. Fig. 4f lacks GAPDH and Fig. 4i total STAT3. In Fig. 4k, it would be better to demonstrate the absolute numbers of both left and right tumors, to appreciate whether the drug also inhibited the non-irradiated tumors. Moreover, the activation status of STAT3 in each of these tumors before and after treatment with Ruxolitinib should be demonstrated.

We agree with the reviewer that it would be useful to include the additional mentioned controls, some of which were performed in the initial submission but not included (ie. GAPDH, Figure 4i). As requested by the reviewer, we now include Western blot images showing total STAT levels (Figure .4c, e, n). In addition, we show the level of p53 proteins in conditioning cells (Figure 4a). To confirm Ras activation we used phospho-ERK antibodies to detect ERK activation downstream of Ras. ERK activity was elevated in MCF cells transfected with oncogenic Ras compared to untransfected controls (Figure. 4b). The remaining cell lines have been previously authenticated and are widely recognized as harboring endogenous oncogenic Ras mutations⁷. Moreover, we now include the GAPDH loading control for what was Figure 4f, now Figure 4j. Furthermore, we present the absolute tumor sizes from the mouse xenograph experiments (Figure 4p) as requested by the reviewer. Finally, we present anti-STAT3 western blots from lysates derived from xenographs tissues demonstrating the effect of Ruxolitinib treatments on STAT levels in the xenographs (Figure 4r).

4. In a mouse model of pancreatic cancer, it was shown that absence of p53 rather than p53 stabilization promotes STAT3 activation in combination with RAS via accumulation of ROS (Wörmann et al., Gastroenterology 151:180-93; 2016). What is the alternative mechanism of non-cell autonomous STAT3 activation upon p53 stabilization in human cells in Figure 4? Which cytokines may be involved?

The reviewer is correct. It is becoming clear that p53 biology is far more complex than previously appreciated. Stabilizing and gain of function p53 mutations phenocopy p53 loss⁸⁻¹². Also, STAT is activated by diverse upstream cellular events, including ROS mediated by loss of p53 function. Here we show that elevated levels of wild-type p53 cooperate with oncogenic Ras to induce nonautonomous STAT activity. Rather than acting through ROS, we propose that p53 binds to and directly stimulates STAT ligands (IL6). Consistent with this, we identified p53 binding sites upstream of *upd* (IL6-like) genes in *Drosophila* (Supplementary Figure 5).

5. According to the data presented in this manuscript, p53 only promotes STAT3 activation in the context of oncogenic RAS. What is the precise contribution of RAS signaling? How does it modify the p53 transcriptional program? Is this effect determined by some threshold of p53 activation? Could other factors be involved?

The cooperation between p53 and oncogenic *Ras* signaling in non-autonomous STAT activation likely relies on the ability of *Ras* signaling to override p53-induced cell death and to elevate *Upd* secretion above a critical threshold.

Ectopic expression of p53 potently triggers apoptosis in otherwise wild-type cells, causing the rapid clearance of these cells. However, oncogenic *Ras* may suppress or delay p53-mediated cell

death, thereby maintaining JAK/STAT cytokine-producing cells in the tumor milieu. Consistent with this, Ras reduced the proportion of IR-induced cell death (see figure right below)

Further, Ras known ability to dramatically stimulate cellular secretion^{13,14} likely elevates the secretion of Upd ligands to the surrounding tissues, leading to robust STAT activities in these cells. In addition to these quantitative effects, Ras signaling may influence the association of p53 with distinct transcription coactivators to favor the expression of specific p53 target genes¹⁵⁻¹⁹. In the context of this study, these target genes may include TGF α cytokines, in addition to Upd²⁰.

6. In Figure 5C it appears that IR by itself is sufficient to induce the unpaired cytokines, and that RAS activation stimulated his effect even further. Is the same observed upon p53 OE (Fig. 3d) or ptp absence (Fig. 3e) alone?

The reviewer is correct. Increasing Wild-type p53 protein levels either via irradiation or ptp absence alone or direct p53 overexpression are sufficient to stimulate Upd cytokines and Ras elevates this effect. The data showing *upd* stimulation following p53 overexpression alone were presented in Supplementary Figure 5 and discussed in the initial submission (now on page 14, lines 321-327). We apologize if this was not clear. The upd stimulation in *ptip*^{-/-} mutant tissues data have been added to Supplementary Figure 5e and to the main text (Page 14, line 315-318).

7. I would propose to change the title to: “Cooperation between oncogenic RAS and wild-type p53 stimulates JAK/STAT non-cell autonomously to promote tumor radioresistance.”, because it could be relevant to mention that wild-type p53 rather than mutant p53 plays a role and “RAS tumor radioresistance” is not a well-defined term. However, this is only a suggestion and may not be followed if the authors don’t agree.

We agree with the reviewer and have changed the title to read: “Cooperation between oncogenic RAS and wild-type p53 stimulates STAT non-cell autonomously to promote tumor radioresistance”

Reviewer #2 (Remarks to the Author): Chabu, Xu and colleagues use a *Drosophila* epithelial model (the eye imaginal disc) to carry out an enhancer/suppressor genetic screening of RasV12-driven tumorigenesis and present evidence that mutations in *ptip1*, which are poorly recovered under normal circumstances, promote non-autonomous tissue growth when combined with RasV12-overexpression. Authors present evidence that these clones have high levels of Dp53 and ectopic expression of Upd cytokines and demonstrate that the Dp53-Upd axis has a major role in RasV12/*ptip*-driven non-autonomous overgrowth.

In this context, it would be necessary to see whether Dp53 is transcriptionally activated (or just the protein stabilized) and whether bona fide transcriptional reporters of Dp53 are being induced.

We performed quantitative PCR and immunostaining experiments to address the reviewer's question. The *ptip*^{-/-} mutation transcriptionally induces *dp53* and *dacapo/p21* (a bona fide p53 target gene) in otherwise wild-type and *Ras*^{v12} tissues. We thank the reviewer for suggesting these experiments as the resulting data have strengthened the manuscript. These data have been included in Figure 2d-g', j, k and are discussed in the main text (page 6, lines 116-118, 123-127).

Also, epistatic relationship to show that Upd expression relies on the activity of Dp53 in RasV12/*ptip* mutant clones should be shown.

We thank the reviewer for suggesting these experiments. We performed *upd* rtPCR experiments on *Ras*^{v12} or *Ras*^{v12}, *ptip*^{-/-} double defective mutant tissues in the presence or absence of dp53 function. The *ptip*^{-/-} mutation stimulated the expression of *upd* cytokines but this effect was lost upon p53 knockdown (p53-RNAi) or expression of the transcriptionally inactive p53 mutant version (p53^{R155H}). These data have been added to Figure 3f and in the main text (page 8, lines 173-176)

Similarly, a role of Upds in RasV12/*ptip*-driven overgrowth should be experimentally validated. In the original submission we showed that depletion of Upd and Upd2 in *Ras*^{v12}, *ptip*^{-/-} clones suppressed the nonautonomous growth phenotype (Page 8, lines 179-183-this version- and Supplementary Fig.2 e, f, k, and l). We apologize if that wasn't clear.

Authors validate their observations in cancer cell lines and xenograft experiments and, in the last part of the ms, authors present evidence that ionizing radiation (known to cause DNA damage) cooperates with RasV12 to promote tissue growth. Nicely, authors demonstrate that Dp53 protein and Upd expression levels are also increased in IR-treated RasV12 tissues and that Dp53 and Upds are required for the growth of RasV12 expressing cells upon IR treatment. The ms is well written, figures are self-explanatory, and experimental data require some epistatic analysis to demonstrate their claim not only in the RasV12/*ptip* condition but also in the RasV12/IR experimental setting. In this regard, experiments concerning the regulation of Upds by Dp53 downstream of Ras/IR and bona fide reporters of Dp53 should be added.

In addition to showing that the *ptip*^{-/-} mutation transcriptionally induces p53 and correspondingly elevate p21 in *Ras*^{v12} clones (Figure 2d-g', j, and k) we found that *ptip*^{-/-} stimulates the expression of Upd cytokines in a p53-dependent manner (Figure 3f). Following the reviewer's suggestion,

we examined the effect of irradiation on p53 and dacapo/p21 expression in *Ras*^{V12} tissues and found that it increases p53 and dacapo levels (Figure 5a-d'). Finally, we had shown that *Ras*^{V12} tissues activates *upd* cytokines (especially *upd2* and *upd3*) in response to irradiation (Figure 5e). We now show that blocking p53 either via RNAi-knockdown or by expressing a dominant-negative version (p53^{R155H}) in irradiated *Ras*^{V12} clones inhibits the upregulation of Upd ligands (Figure 5f). These findings have been added to the main text (page 11, lines 250-253)

Unfortunately, the major message of the ms - as stated in the title, abstract, results and discussion on the role of Dp53 and STAT depletion in radio-sensitizing RAS tumors - is not substantiated by the data. Authors present evidence that Dp53 and STAT are required for the growth of IR-treated RAS clones (most probably as a consequence of the well-known mitogenic activity of Upds). Authors should present data on cell death to demonstrate that depletion of Dp53 and STAT sensitizes RAS-clones to IR-induced cell death.

There seem to be a misunderstanding. As stated in the original submission, we define radio-resistance as incomplete sensitivity to radiotherapy or cancers' capacity to rapidly reform following radiation therapy (initial submission: page 9, lines 219-220 and page 10, lines 221-223). This was also highlighted in the original abstract (page 2, lines 32-33) and in the discussion (page 14, lines 336-339). The Ras/p53-STAT signaling axis drives tumor recurrence following radiotherapy mainly by accelerating Ras tumor growth. Consistent with this, we detected a considerable STAT-dependent increase of cell proliferation when Ras cells were cultured in media conditioned with P53-overexpressing or irradiated cells (Supplementary Figure 3 and 4). STAT inhibition had only limited to no effect on cell death in human cells (Supplementary Figure 3, 4), In flies, blocking STAT by expressing a validated dominant-negative version of the STAT receptor Domeless (*Dome*^{DN}) increased cell death by only ~2%, compared to wild-type cells which show well over a five-fold increase in cell death in response to irradiation (see figure immediately below). We apologize for not making this distinction clearer. To avoid any potential confusion from readers, we now clearly define radioresistance much earlier in the introduction (page 3, lines 46-47).

Reviewer #3 (Remarks to the Author): The authors of the manuscript posit a connection between genotoxic stress and oncogenic RAS activation leading to cell proliferation in the tumor microenvironment via a p53 mediated JAK/STAT activation pathway. Interestingly, they argue that these results can explain the paradoxical tumor radioresistance in Ras tumours with wt p53, and that a possible avenue of therapy rescue may be co-treatment with JAK/STAT inhibitors. The authors identify this mechanism in an animal model using *Drosophila* genetic screening, where a uniquely genotoxic mutation *ptip*^{-/-} acts to destabilize the genome, but also promote overgrowth of surrounding wildtype tissue via p53. The approach is interesting, as using animal models in this way may reveal complex cell-cell interactions dependant on a complete tissue microenvironment otherwise lacking in normal cell cultures. The authors further confirm this finding by specific gene regulation studies using qPCR, as well as human breast and lung cancer monolayer cell cultures and in vivo mouse xenograft models together with ionizing radiation for genotoxicity. Importantly, the effect persisted in all model systems across species, indicating a fundamental and evolutionary preserved mechanism of action. Most significantly, existing drugs targeting STAT signaling could inhibit the nonautonomous growth-inducing effect of p53 and ionizing radiation in cell cultures (Ganetespib) and mice (Ruxolitinib). The authors lastly closely examine the cellular signaling pathways and elucidate the uniqueness of the described interaction, in comparison to e.g. Salvador/Sav etc.

All in all, the presented work is a thorough and convincing body of evidence of the connection between oncogenic Ras mutations and cell proliferation in combination with genotoxic treatments such as ionizing radiation, mediated by p53 and JAK/STAT induction.

* The paper is technically sound

* The claims are convincing

* The claims are on the whole supported by the experimental data presented

* The statistical analysis of the data is sound

* The claims are appropriately discussed in the context of previous literature

Minor concerns:

As the authors know, the investigated p53 levels are finely regulated by members of the MDM-family, like MDM2 and MDMX. What are the levels of these proteins and does their expression affect radio-sensitivity and apoptotic signalling? At what point do proteins of the negative feedback loop peak and what role do they play in p53 regulation in the model presented? Ras signaling transcriptionally induces MDM2^{21,22}, thus we would expect that MDM2 levels will be elevated in Ras^{V12} cells compared to wild-type controls. Irradiation will likely further elevate MDM2 levels while reducing MDMX. Under DNA damage ATM phosphorylates and targets MDMX for degradation via MDM2²³. In DNA damage settings ATM acts via CHK1/2 to activate p53 and to protect it from MDM2-mediated degradation, leading to increased p53 levels²⁴. Inhibiting ATM would reduce p53 levels²⁵. Consistent with the reviewer's suggestion, we knocked down ATM proteins (ATM3 and ATM6) in our *Drosophila* Ras tumor irradiation model and found that, similar to p53 inhibition, this sensitizes Ras tissues to irradiation treatment (Figure below a, a' versus c, c' or e, e'; relative GFP-positive area). These data provide further evidence that the P53 pathway plays a key role in Ras tumor radioresistance. We define

radioresistance as incomplete sensitivity to radiotherapy or the capacity for irradiated cells to rapidly grow and reform tumors following radiation therapy.

Reference

- 1 Munoz, I. M., Jowsey, P. A., Toth, R. & Rouse, J. Phospho-epitope binding by the BRCT domains of hPTIP controls multiple aspects of the cellular response to DNA damage. *Nucleic Acids Res* **35**, 5312-5322, doi:10.1093/nar/gkm493 (2007).
- 2 Jowsey, P. A., Doherty, A. J. & Rouse, J. Human PTIP facilitates ATM-mediated activation of p53 and promotes cellular resistance to ionizing radiation. *J Biol Chem* **279**, 55562-55569, doi:10.1074/jbc.M411021200 (2004).
- 3 Gong, Z., Cho, Y. W., Kim, J. E., Ge, K. & Chen, J. Accumulation of Pax2 transactivation domain interaction protein (PTIP) at sites of DNA breaks via RNF8-dependent pathway is required for cell survival after DNA damage. *J Biol Chem* **284**, 7284-7293, doi:10.1074/jbc.M809158200 (2009).
- 4 Wu, Q. *et al.* In vivo CRISPR screening unveils histone demethylase UTX as an important epigenetic regulator in lung tumorigenesis. *Proc Natl Acad Sci U S A* **115**, E3978-E3986, doi:10.1073/pnas.1716589115 (2018).
- 5 Nakamura, M., Ohsawa, S. & Igaki, T. Mitochondrial defects trigger proliferation of neighbouring cells via a senescence-associated secretory phenotype in *Drosophila*. *Nat Commun* **5**, 5264, doi:10.1038/ncomms6264 (2014).
- 6 Murcia, L., Clemente-Ruiz, M., Pierre-Elies, P., Royou, A. & Milan, M. Selective Killing of RAS-Malignant Tissues by Exploiting Oncogene-Induced DNA Damage. *Cell Rep* **28**, 119-131 e114, doi:10.1016/j.celrep.2019.06.004 (2019).
- 7 Blanco, R. *et al.* A gene-alteration profile of human lung cancer cell lines. *Hum Mutat* **30**, 1199-1206, doi:10.1002/humu.21028 (2009).
- 8 Oren, M. & Rotter, V. Mutant p53 gain-of-function in cancer. *Cold Spring Harb Perspect Biol* **2**, a001107, doi:10.1101/cshperspect.a001107 (2010).
- 9 Freed-Pastor, W. A. & Prives, C. Mutant p53: one name, many proteins. *Genes & development* **26**, 1268-1286, doi:10.1101/gad.190678.112 (2012).

- 10 Muller, P. A. & Vousden, K. H. p53 mutations in cancer. *Nat Cell Biol* **15**, 2-8, doi:10.1038/ncb2641 (2013).
- 11 Brosh, R. & Rotter, V. When mutants gain new powers: news from the mutant p53 field. *Nat Rev Cancer* **9**, 701-713, doi:10.1038/nrc2693 (2009).
- 12 Terzian, T. *et al.* The inherent instability of mutant p53 is alleviated by Mdm2 or p16INK4a loss. *Genes & development* **22**, 1337-1344, doi:10.1101/gad.1662908 (2008).
- 13 Chabu, C. & Xu, T. Oncogenic Ras stimulates Eiger/TNF exocytosis to promote growth. *Development* **141**, 4729-4739, doi:10.1242/dev.108092 (2014).
- 14 Issaq, S. H., Lim, K. H. & Counter, C. M. Sec5 and Exo84 foster oncogenic ras-mediated tumorigenesis. *Mol Cancer Res* **8**, 223-231, doi:10.1158/1541-7786.MCR-09-0189 (2010).
- 15 Zhao, R. *et al.* Analysis of p53-regulated gene expression patterns using oligonucleotide arrays. *Genes & development* **14**, 981-993 (2000).
- 16 Kapoor, M. & Lozano, G. Functional activation of p53 via phosphorylation following DNA damage by UV but not gamma radiation. *Proc Natl Acad Sci U S A* **95**, 2834-2837, doi:10.1073/pnas.95.6.2834 (1998).
- 17 Lu, H., Taya, Y., Ikeda, M. & Levine, A. J. Ultraviolet radiation, but not gamma radiation or etoposide-induced DNA damage, results in the phosphorylation of the murine p53 protein at serine-389. *Proc Natl Acad Sci U S A* **95**, 6399-6402, doi:10.1073/pnas.95.11.6399 (1998).
- 18 Webley, K. *et al.* Posttranslational modifications of p53 in replicative senescence overlapping but distinct from those induced by DNA damage. *Mol Cell Biol* **20**, 2803-2808, doi:10.1128/mcb.20.8.2803-2808.2000 (2000).
- 19 Oda, K. *et al.* p53AIP1, a potential mediator of p53-dependent apoptosis, and its regulation by Ser-46-phosphorylated p53. *Cell* **102**, 849-862, doi:10.1016/s0092-8674(00)00073-8 (2000).
- 20 Hobor, S. *et al.* TGFalpha and amphiregulin paracrine network promotes resistance to EGFR blockade in colorectal cancer cells. *Clin Cancer Res* **20**, 6429-6438, doi:1078-0432.CCR-14-0774 [pii] 10.1158/1078-0432.CCR-14-0774 (2014).
- 21 Ries, S. *et al.* Opposing effects of Ras on p53: transcriptional activation of mdm2 and induction of p19ARF. *Cell* **103**, 321-330, doi:10.1016/s0092-8674(00)00123-9 (2000).
- 22 Halaschek-Wiener, J., Wacheck, V., Kloog, Y. & Jansen, B. Ras inhibition leads to transcriptional activation of p53 and down-regulation of Mdm2: two mechanisms that cooperatively increase p53 function in colon cancer cells. *Cell Signal* **16**, 1319-1327, doi:10.1016/j.cellsig.2004.04.003 (2004).
- 23 Chen, L., Gilkes, D. M., Pan, Y., Lane, W. S. & Chen, J. ATM and Chk2-dependent phosphorylation of MDMX contribute to p53 activation after DNA damage. *EMBO J* **24**, 3411-3422, doi:10.1038/sj.emboj.7600812 (2005).
- 24 Bartek, J. & Lukas, J. Chk1 and Chk2 kinases in checkpoint control and cancer. *Cancer Cell* **3**, 421-429, doi:10.1016/s1535-6108(03)00110-7 (2003).
- 25 Jiang, H. *et al.* The combined status of ATM and p53 link tumor development with therapeutic response. *Genes & development* **23**, 1895-1909, doi:10.1101/gad.1815309 (2009).

REVIEWERS' COMMENTS:

Reviewer #1 (Remarks to the Author):

I have no more comments, all issues have been addressed sufficiently.

Reviewer #3 (Remarks to the Author):

No further questions.

Response to reviewers' comments

Manuscript#: COMMSBIO-20-0984-T

Title: Cooperation between oncogenic Ras and Wild-type p53 stimulates JAK/STAT non-cell autonomously to promote Ras tumor radioresistance"

The referees had no further comments or concerns to be addressed.